# SURPRISE MINIMIZING MULTI-AGENT LEARNING WITH ENERGY-BASED MODELS

## ABSTRACT

Multi-Agent Reinforcement Learning (MARL) has demonstrated significant success by virtue of collaboration across agents. Recent work, on the other hand, introduces *surprise* which quantifies the degree of change in an agent's environment. Surprise-based learning has received significant attention in the case of single-agent entropic settings but remains an open problem for fast-paced dynamics in multi-agent scenarios. A potential alternative to address surprise may be realized through the lens of free-energy minimization. We explore surprise minimization in multi-agent learning by utilizing the free energy across all agents in a multi-agent system. A temporal Energy-Based Model (EBM) represents an estimate of surprise which is minimized over the joint agent distribution. Our formulation of the EBM is theoretically akin to the minimum conjugate entropy objective and highlights suitable convergence towards minimum surprising states. We further validate our theoretical claims in an empirical study of multi-agent tasks demanding collaboration in the presence of fast-paced dynamics.

sites.google.com/view/surprise-web/

## 1 INTRODUCTION

The rise of RL has led to an increasing interest in the study of multi-agent systems (Lowe et al., 2017; Vinyals et al., 2019), commonly known as Multi-Agent Reinforcement Learning (MARL). In the case of partially observable settings, MARL enables the learning of policies with centralised training and decentralised control (Kraemer & Banerjee, 2016). This has proven to be useful for exploiting value-based methods which motivate collaboration across large number of agents. But how do agents behave in the presence of sudden environmental changes?

Consider the problem of autonomous driving wherein a *driver* (agent) autonomously operates a vehicle in real-time. The *driver* learns to optimize the reward function by maintaining constant speed and covering more distance in different traffic conditions. Whenever the vehicle approaches an obstacle, the *driver* acts to avoid it by utilizing the brake and directional steering commands. However, due to the fast-paced dynamics of the environment, say fast-moving traffic, the agent may abruptly encounter an obstacle (*a person running across the street*) which may result in a collision. Irrespective of the optimal action (*pushing of brakes*) executed by the agent, the vehicle may fail to evade the collision as a result of the abrupt temporal change.

The above arises as a consequence of *surprise*, which is defined as a statistical measure of uncertainty. Surprise minimization (Berseth et al., 2019) is a recent phenomenon observed in the case of single-agent RL methods which deals with environments consisting of rapidly changing states. In the case of model-based RL (Kaiser et al., 2019), surprise minimization is used as an effective planning tool in the agent's model (Berseth et al., 2019) whereas in the case of model-free RL, surprise minimization is witnessed as an intrinsic motivation (Achiam & Sastry, 2017; Macedo et al., 2004) or generalization problem (Chen, 2020). On the other hand, MARL does not account for surprise across agents as a result of which agents remain unaware of drastic changes in the environment (Macedo & Cardoso, 2005). Thus, surprise minimization in the case of multi-agent settings requires attention from a critical standpoint.

A potential pathway to treat surprising states may be realized in light of free-energy minimization. The free-energy principle depicts convergence to local niches and provides a general recipe for cognitive stability among agents. Through this lens, we unify surprise with free-energy in the multi-agent setting. We construct a temporal EBM which represents an estimate of surprise agents

may face in the environment. All agents jointly minimize this estimate utilizing temporal difference learning upon their value functions and the EBM. Our formulation of free-energy minimization is theoretically akin to minimizing the entropy in conjugate gradient space. This insight provides a suitable convergence result towards minimum surprising states (or niches) of the agent state distributions. In an empirical study of multi-agent tasks which present significant collaboration bottlenecks and fast-paced dynamics, we validate our theoretical claims and motivate the practical usage of EBMs in MARL.

## 2 RELATED WORK

**Surprise Minimization:** Despite the recent success of value-based methods (Mnih et al., 2016; Hessel et al., 2017) RL agents suffer from spurious state spaces and encounter sudden changes in trajectories. Quantitatively, surprise has been studied as a measure of deviation (Berseth et al., 2019; Chen, 2020) among states encountered by the agent during its interaction with the environment. While exploring (Burda et al., 2019; Thrun, 1992) the environment, agents tend to have higher deviation among states which is gradually reduced by gaining a significant understanding of state-action transitions. In the case of model-based RL, agents can leverage spurious experiences (Berseth et al., 2019) and plan effectively for future steps. On the other hand, in the case of model-free RL, surprise results in sample-inefficient learning (Achiam & Sastry, 2017). This is primarily addressed by making use of rigorous exploration strategies (Stadie et al., 2015; Lee et al., 2019). High-dimensional exploration further requires extrinsic feature engineering (Kulkarni et al., 2016) and meta models (Gupta et al., 2018). A suitable way to tackle high-dimensional dynamics is by utilizing surprise as a penalty on the reward (Chen, 2020). This leads to improved generalization for single-agent interactions (Ren et al., 2005). Our proposed approach is orthogonal to the aforesaid methods.

**Energy-based Models:** EBMs have been successfully implemented in single-agent RL methods (O'Donoghue et al., 2016; Haarnoja et al., 2017). These typically make use of Boltzmann distributions to approximate policies (Levine & Abbeel, 2014). Such a formulation results in the minimization of free energy within the agent. While policy approximation depicts promise in the case of unknown dynamics, inference methods (Toussaint, 2009) play a key role in optimizing goal-oriented behavior.

A second type of usage of EBMs follows the maximization of entropy (Ziebart et al., 2008). The maximum entropy framework (Haarnoja et al., 2018b) highlighted in Soft Q-Learning (SQL) (Haarnoja et al., 2017) allows the agent to obey a policy which maximizes its reward and entropy concurrently. Maximization of agent's entropy results in diverse and adaptive behaviors (Ziebart, 2010) which may be difficult to accomplish using standard exploration techniques (Burda et al., 2019; Thrun, 1992). The maximum entropy framework is akin to approximate inference in the case of policy gradient methods (Schulman et al., 2017). Such a connection between likelihood ratio gradient techniques and energy-based formulations leads to diverse and robust policies (Haarnoja, 2018) and their hierarchical extensions (Haarnoja et al., 2018a) which preserve the lower levels of hierarchies. In the case of MARL, EBMs have witnessed limited applicability as a result of the increasing number of agents and complexity within each agent (Buşoniu et al., 2010). While the probabilistic framework is readily transferable to opponent-aware multi-agent systems (Wen et al., 2019), cooperative settings consisting of coordination between agents require a firm formulation of energy which is scalable in the number of agents (Grau-Moya et al., 2018) and accounts for environments consisting of spurious states (Wei et al., 2018). Our theoretical formulation is motivated by these methods in literature.

## 3 PRELIMINARIES

### 3.1 MULTI-AGENT LEARNING

We review the cooperative MARL setup. The problem is modeled as a Dec-Partially Observable Markov Decision Process (POMDP) (Oliehoek & Amato, 2016) defined by the tuple $(\mathcal{S}, \mathcal{A}, r, N, P, Z, O, \gamma)$ where the state space $\mathcal{S}$ and action space $\mathcal{A}$ are discrete, $r : \mathcal{S} \times \mathcal{A} \to [r_{min}, r_{max}]$ presents the reward observed by agents $a \in N$ where $N$ is the set of all agents, $P : \mathcal{S} \times \mathcal{S} \times \mathcal{A} \to [0, \infty)$ presents the unknown transition model consisting of the transition probability to the next state $s' \in \mathcal{S}$ given the current state $s \in \mathcal{S}$ and joint action $u \in \mathcal{A}$ (a combination of each agent's action $u^a \in \mathcal{A}^a$) at time step $t$ and $\gamma$ is the discount factor. We consider a partially observable setting in which each agent $n$ draws individual observations $z \in Z$ according to the observation function $O(s, u) : \mathcal{S} \times \mathcal{A} \to Z$. We consider a joint policy $\pi_\theta(u|s)$ as a function of

model parameters $\theta$. Standard RL defines the agent's objective to maximize the expected discounted reward $\mathbb{E}_{\pi_\theta}[\sum_{t=0}^{T} \gamma^t r(s_t, u_t)]$ as a function of the parameters $\theta$. The joint action-value function for agents is represented as $Q(u, s; \theta) = \mathbb{E}_{\pi_\theta}[\sum_{t=1}^{T} \gamma^t r(s, u)|s = s_t, u = u_t]$ which is the expected sum of payoffs obtained in state $s$ upon performing action $u$ by following the policy $\pi_\theta$. We denote the optimal policy $\pi_{\theta*}$ ( shorthand $\pi^*$) such that $Q(u, s; \theta^*) \geq Q(u, s; \theta) \forall s \in S, u \in A$. In the case of multiple agents, the joint optimal policy can be expressed as the Nash Equilibrium (Nash, 1950) of the Stochastic Markov Game as $\pi^* = (\pi^{1,*}, \pi^{2,*}, ...\pi^{N,*})$ such that $Q(u^a, s; \theta^*) \geq Q(u^a, s; \theta) \forall s \in S, u \in A, a \in N$. Q-Learning is an off-policy, model-free algorithm suitable for continuous and episodic tasks. The algorithm uses semi-gradient descent to minimize the Temporal Difference (TD) error in Equation 1.

$$\mathbb{L}(\theta) = \mathbb{E}_{s,u,s'\sim\mathcal{R}} \left[ \left( r + \gamma \max_{u'\in A} Q(u', s'; \theta^-) - Q(u, s; \theta) \right)^2 \right] \tag{1}$$

where $y = r + \gamma \max_{u'\in A} Q(u', s'; \theta^-)$ is the TD target consisting of $\theta^-$ as the target parameters and $\mathcal{R}$ denotes the replay buffer.

### 3.2 ENERGY-BASED MODELS

EBMs (LeCun et al., 2006; 2007) have been successfully applied in the field of machine learning (Teh et al., 2003) and probabilistic inference (MacKay, 2002). A typical EBM $\mathcal{E}$ formulates the equilibrium probabilities (Sallans & Hinton, 2004) $P(v, h) = \frac{\exp(-\mathcal{E}(v,h))}{\sum_{\hat{v},\hat{h}}[\exp(-\mathcal{E}(\hat{v},\hat{h}))]}$ via a Boltzmann distribution (Levine & Abbeel, 2014) where $v$ and $h$ are the values of the visible and hidden variables and $\hat{v}$ and $\hat{h}$ are all the possible configurations of the visible and hidden variables respectively. The probability distribution over all the visible variables can be obtained by summing over all possible configurations of the hidden variables. This is mathematically expressed in Equation 2.

$$P(v) = \frac{\sum_h \exp(-\mathcal{E}(v, h))}{\sum_{\hat{v},\hat{h}} \exp(-\mathcal{E}(\hat{v}, \hat{h}))} \tag{2}$$

Here, $\mathcal{E}(v, h)$ is called the equilibrium free energy which is the minimum of the variational free energy and $\sum_{\hat{v},\hat{h}} \exp(-\mathcal{E}(\hat{v}, \hat{h}))$ is the partition function.

## 4 ENERGY-BASED SURPRISE MINIMIZATION

We begin by constructing surprise minimization as an energy-based problem in the temporal setting. The motivation behind an energy-based formulation stems from rapidly changing states as an undesired niche among agents in the case of partially-observed settings. To steer agents away from this niche, we further construct a method which incorporates the theoretical aspect of the study.

### 4.1 THE SURPRISE MINIMIZATION OBJECTIVE

To make analysis tractable towards valid function spaces and surprising states, we take into account two assumptions which form the central basis of surprise minimization among multiple agents.

> **Assumption 1.** (Completeness of value function space) *The space $\Pi : \mathcal{S} \times \mathcal{A}$ of all Q value functions $Q(s, u) \in \Pi, \; \forall s \in \mathcal{S}, \; \forall u \in \mathcal{A}$ is a nonempty complete metric space.*

Assumption 1 restricts the formulation of individual agent value functions $Q_a$ to the nonempty complete metric space. A nonempty space confirms the presence of candidate functions $Q_a$ upper bounded by the optimal function $Q^*$, i.e.- $Q_a \leq Q^*, \; \forall a \in N$ (Bertsekas & Tsitsiklis, 1995). The completeness counterpart, on the other hand, provisions a fixed interior **int** $\Pi$ for optimization (Boyd & Vandenberghe, 2004).

> **Assumption 2.** (Constant surprise at Equilibrium) *In the limit of convergence* $\lim_{\pi_a \to \pi^*}$ *to an optimal policy* $\pi^*$, *all agents* $a \in N$ *incur a finite surprise* $\zeta > 0$ *between consecutive states $s$ and $s'$ until termination state $s_T$.*

Assumption 2 is directly based on the constant and continuous temporal aspect of surprise minimization (Schwartenbeck et al., 2013; Friston, 2010). Corresponding to the lifetime of each agent $a \in N$, a desired ecological niche bakes in the optimal distribution of actions which correspond to minimum yet finite instantaneous surprise.

We formulate the energy-based objective consisting of surprise as a function of states $s$, joint actions $u$ and standard deviation $\sigma$ of observations for each agent $a$. In the case of high-dimensional state spaces (such as multiple opponents), $\sigma$ informs agents of the abrupt statistical change that would take place upon executing action $u$. We formulate surprise as $\mathcal{T}V_{\mathrm{surp}}^a(s, u, \sigma)$ which serves as an uncertainty quantifier Unc(s,a) of the state-action distribution. Here $V_{\mathrm{surp}}^a(s, u, \sigma)$ denotes the surprise value function which serves as a mapping from agent and environment dynamics to surprise. Define an operator presented in Equation 3 which sums surprising configurations across all agents.

$$\mathcal{T}V_{\mathrm{surp}}^a(s, u, \sigma) = \log \sum_{a=1}^{N} \exp\left(V_{\mathrm{surp}}^a(s, u, \sigma)\right) \tag{3}$$

**Remark 1.** $\mathcal{T}V_{\mathrm{surp}}^a(s, u, \sigma)$ *intuitively provides a global estimate of surprise. If all agents are equally likely to face a surprising state, then* $\mathcal{T}V_{\mathrm{surp}}^a(s, u, \sigma)$ *captures their individual contributions.*

The formulation makes use of the soft-maximum operator (Asadi & Littman, 2017). The operator $\mathcal{T}V_{\mathrm{surp}}^a(s, u, \sigma)$ is similar to prior energy formulations (Haarnoja et al., 2017) where the energy across different actions is evaluated. In our case, inference is carried out across all agents with actions as prior variables. However, in the special case of using an EBM as a $Q$-function, our approach suitable generalizes to the above methods (details in Appendix B).

Our choice of $\mathcal{T}V_{\mathrm{surp}}^a(s, u, \sigma)$ is based on its unique mathematical properties which result in better convergence. Of these properties, the most useful result is that $\mathcal{T}$ forms a contraction on the surprise value function $V_{\mathrm{surp}}^a(s, u, \sigma)$ indicating a guaranteed minimization of surprise within agents. This is formally stated in Theorem 1 while utilizing the completeness criterion of Assumption 1 which provides a tractable value function space. All proofs are deferred to Appendix A.

> **Theorem 1.** *Given a surprise value function* $V_{\mathrm{surp}}^a(s, u, \sigma) \ \forall a \in N$, *the energy operator* $\mathcal{T}V_{\mathrm{surp}}^a(s, u, \sigma) = \log \sum_{a=1}^{N} \exp\left(V_{\mathrm{surp}}^a(s, u, \sigma)\right)$ *forms a contraction on* $V_{\mathrm{surp}}^a(s, u, \sigma)$.

Theorem 1 provides a suitable guarantee of $\mathcal{T}V_{\mathrm{surp}}^a(s, u, \sigma)$ converging to a fixed point niche. The contraction result is directly based on Banach's fixed point property and suggests the generalization of convergence in any nonempty complete metric space $(X, d)$ (Bertsekas & Tsitsiklis, 1995).

We now consider a weighted combination of $Q(s, u)$ with $\mathcal{T}V_{\mathrm{surp}}^a(s, u, \sigma)$ wherein we denote $\beta$ as a temperature parameter,

$$\hat{Q}(u, s; \theta) = Q(u, s; \theta) + \beta \log \sum_{a=1}^{N} \exp\left(V_{\mathrm{surp}}^a(s, u, \sigma)\right)) \tag{4}$$

**Remark 2.** *Equation 4 is an instance of value function regularization wherein the Q values are subject to a joint penalty while observing surprising states.*

Interestingly, upon considering the Legendre transform $f^*(x)$ (Boyd & Vandenberghe, 2004; Gao & Pavel, 2017) (convex conjugate function corresponding to the conjugate space $\mathcal{X}$ of a differentiable function $f(\mathrm{z})$) of $\mathcal{T}V_{\mathrm{surp}}^a(s, u, \sigma)$, we obtain the following,

$$f^*(x) = \sup_{\mathrm{z} \in \mathrm{dom} f} \left(x^{\mathrm{T}}\mathrm{z} - f(\mathrm{z})\right) , \ f(\mathrm{z}) = \mathcal{T}V_{\mathrm{surp}}^a(s, u, \sigma) \tag{5}$$

$$f^*(x) = \sum_x x \log(x) , \ x = \nabla_{\mathrm{z}} f(\mathrm{z}) \in \mathcal{X} \tag{6}$$

**Remark 3.** *The Legendre Transform of* $\mathcal{T}V_{\mathrm{surp}}^a(s, u, \sigma)$ *given by* $f^*(x) = \sum_x x \log(x)$ *when utilized as value function regularization* $\hat{Q} = Q - f^*(x)$ *corresponds to the minimum entropy formulation in conjugate space* $\mathbb{E}_{\pi_\theta}\left[\sum_{t=0}^{T} \gamma^t (r(s_t, u_t) - \lambda \mathcal{H}(x))\right]$ *for* $x = \nabla_{\mathrm{z}} f(\mathrm{z}) \in \mathcal{X}$.

Based on the above insight, minimizing entropy to express $\nabla_z f(z)$ in conjugate space is akin to minimizing uncertainty among all agents in the value function space $\Pi$. Intuitively, $\mathcal{H}(x)$ denotes the uncertainty for each agent $a \in N$ in the multi-agent population which is directly related to its ability of efficaciously interpreting the environment. Minimizing $\mathcal{H}(x)$ leads to an increase in the expressiveness of value function. This in turn, induces an expressive state visitation distribution which steers the agent away from sudden changes in its environment. Note that the setting does not minimize entropy in value function space which would stand contrary to the maximum entropy formulation Haarnoja et al. (2018b) (see Appendix B).

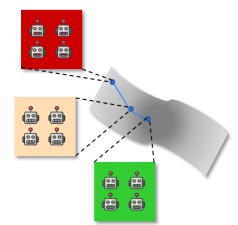

Figure 1: Agent populations (robots) traverse the energy landscape (in grey) during update steps (●) to seek energy minima (darker shade at center). This results in surprise minimization from high (■) to low energy (■) niches.

Figure 1 presents an intuitive illustration of the objective. The joint agent population aims to minimize surprise corresponding to minimum energy configurations. Agents collaborate in partially-observed worlds to attain a joint niche. This local niche implicitly corresponds to a fixed point of $\mathcal{T}V_{\text{surp}}^a(s, u, \sigma)$ on the energy landscape. Note that agents act locally with actions conditioned on their own action observation histories. It is by virtue of preconditioned values estimations that the surprise minimization scheme informs agents of joint surprise. Upon population's convergence to a suitable configuration, agents continue to experience minimum (yet finite) surprise arising from evironment dynamics.

## 4.2 SURPRISE MINIMIZATION WITH FUNCTION APPROXIMATION

We utilize the above insights as surprise-based regularization in the TD learning setting. Upon replacing $Q(u, s; \theta)$ with $\hat{Q}(u, s, ; \theta)$ in the RL construction of Equation 1 one obtains the following,

$$L(\theta) = \mathop{\mathbb{E}}_{s,u,s'\sim\mathcal{R}} \left[ \frac{1}{2} \left( \hat{y} - (Q(u, s; \theta) + \beta \log \sum_{a=1}^{N} \exp\left(V_{\text{surp}}^a(s, u, \sigma)\right)) \right)^2 \right]$$

where $\hat{y} = r + \gamma \max_{u'} Q(u', s'; \theta^-) + \beta \log \sum_{a=1}^{N} \exp\left(V_{\text{surp}}^a(s', u', \sigma')\right)$. Collecting the log terms yields the following,

$$= \mathop{\mathbb{E}}_{s,u,s'\sim\mathcal{R}} \left[ \frac{1}{2} \left( r + \gamma \max_{u'} Q(u', s'; \theta^-) + \beta \log \left( \frac{\sum_{a=1}^{N} \exp\left(V_{\text{surp}}^a(s', u', \sigma')\right)}{\sum_{a=1}^{N} \exp\left(V_{\text{surp}}^a(s, u, \sigma)\right)} \right) - Q(u, s; \theta) \right)^2 \right]$$

$$L(\theta) = \mathop{\mathbb{E}}_{s,u,s'\sim\mathcal{R}} \left[ \frac{1}{2} \left( r + \gamma \max_{u'} Q(u', s'; \theta^-) + \beta E - Q(u, s; \theta) \right)^2 \right] \tag{7}$$

Here, $E$ is defined as the *surprise ratio*. The surprise value function $V_{\text{surp}}^a(s', u', \sigma')$ is expressed as the negative free energy and $\sum_{a=1}^{N} \exp\left(V_{\text{surp}}^a(s, u, \sigma)\right)$ as the partition function of a conventional EBM described in Equation 2. Alternatively, $V_{\text{surp}}^a(s, u, \sigma)$ can be formulated as the negative free energy with $\sum_{a=1}^{N} \exp\left(V_{\text{surp}}^a(s', u', \sigma')\right)$ as the partition function. The TD objective incorporates the minimization of surprise across all agents as minimizing the energy in spurious states.

**Remark 4.** *The above formulation of $\beta E$ can be realized as intrinsic motivation steering the agent towards subgoals with reduced surprise.*

The energy formulation $E$ provides a tractable distribution over all surprising configurations in the state space $\mathcal{S}$. This guarantees convergence to minimum surprise at optimal policy $\pi^*$ and is formally expressed in Theorem 2 (see Appendix C for a detailed convergence analysis).

**Theorem 2.** *Upon agent's convergence to an optimal policy $\pi^*$, total energy of $\pi^*$, expressed by $E^*$ will reach a thermal equilibrium consisting of minimum surprise among consecutive states $s$ and $s'$.*

Theorem 2 demonstrates an intuitive convergence result of agent populations collaborating to reside in a mutual ecological niche (Friston, 2010). The multi-agent population with minimum surprise exhibits the optimal policy $\pi^*$ which results in minimum energy corresponding to each surprising state in the state distribution $\mathcal{S}$. Orthogonally, agents may continue to experience finite and constant surprise in the long-horizon while acting optimally to visit non-surprising and rewarding states. This presents surprise minimization as a secondary surrogate objective in MARL.

## 4.3 ENERGY-BASED MIXER (EMIX)

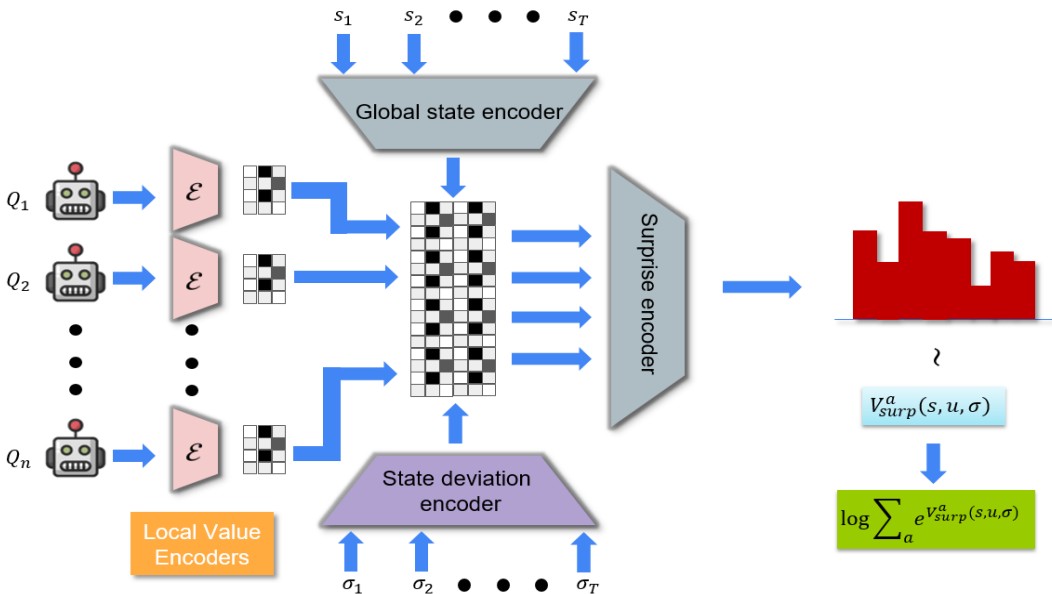

Figure 2: The EMIX architecture for learning surprise across global states.

Based on our theoretical analysis, we incorporate learning of surprise as global intrinsic motivation across all agents in the multi-agent system. A global estimate of surprise, following the energy operator $\mathcal{T}V_{\text{surp}}^a(s, u, \sigma)$, is befitting from a computational perspective as well. An individual estimate of surprise for each agent may be intractable to obtain due to the non-stationarity of the environment. Instead, we seek to minimize surprise jointly across all agents using an expressive Energy-based MIXer (EMIX) architecture which is compatible with any multi-agent RL algorithm. Figure 2 illustrates our learning scheme.

Learning of surprise in the high-dimensional value function space is cumbersome with the number of actions scaling linearly in the number of agents. This imposes an inherent restriction to learn global surprise efficaciously across all agents at a given timestep. Towards this goal, EMIX encodes individual value functions $Q_1$, $Q_2$, ... $Q_n$ corresponding to each agent using local value encoders. These encoders capture the local change in value functions arising over subsequent TD learning iterations (Wang et al., 2021). A global state encoder maps environment states $s_1$, $s_2$, ... $s_T$ to a low dimensional representation. Further, a state deviation encoder encodes deviations across all states $s_1$, $s_2$, ... $s_T$ within the given batch. Akin to a model-based method (Janner et al., 2019), the state deviation encoder accounts for uncertainty in an agent's state visitation distribution. Note that the encoder does not construct an explicit model of states, but only represents their variation in the agent's environment. This insight is essential to account for abrupt dynamics encountered by agents. Representations obtained from state and value function encoders are concatenated and compressed using a final surprise encoder which estimates a distribution of surprise values. The distribution implicitly represents the density of states wherein an agent may encounter most surprise. A value estimate $V_{\text{surp}}^a(s, u, \sigma)$ sampled from the surprise distribution depicts the variational free energy configuration upon application of $\mathcal{T}$ which serves as global intrinsic motivation. Practical training of EMIX proceeds with backpropagation (Rumelhart et al., 1986) using gradient descent and the reparameterization trick (Kingma & Welling, 2014) for sampling of $V_{\text{surp}}^a(s, u, \sigma)$.

## 4.4 PRACTICAL IMPLEMENTATION

---

**Algorithm 1** Energy-based MIXer (EMIX)

---

1: Initialize $\phi, \theta, \theta_1^- ..., \theta_m^-$, agent and hypernetwork parameters.
2: Initialize learning rate $\alpha$, temperature $\beta$ and replay buffer $\mathcal{R}$.
3: **for** environment step **do**
4:     $u \leftarrow (u_1, u_2..., u_N)$
5:     $\mathcal{R} \leftarrow \mathcal{R} \cup \{(s, u, r, s')\}$
6:     **if** $|\mathcal{R}| >$ batch-size **then**
7:         **for** random batch **do**
8:             $Q_{tot}^\theta \leftarrow$ *Mixer-Network*$(Q_1, Q_2..., Q_N, s)$
9:             $Q_i^{\theta^-} \leftarrow$ *Target-Mixer*$_i(Q_1, Q_2..., Q_N, s'), \forall i = 1, 2.., m$
10:           Calculate $\sigma$ and $\sigma'$ using $s$ and $s'$
11:           $V_{\text{surp}}^a(s, u, \sigma) \leftarrow$ *Surprise-Mixer(s, u, σ)*
12:           $V_{\text{surp}}^a(s', u', \sigma') \leftarrow$ *Target-Surprise-Mixer(s', u', σ')*
13:           $E \leftarrow \log \left( \dfrac{\sum_{a=1}^N \exp\left(V_{\text{surp}}^a(s', u', \sigma')\right)}{\sum_{a=1}^N \exp\left(V_{\text{surp}}^a(s, u, \sigma)\right)} \right)$
14:           Calculate $L(\theta)$ using $E$ in Equation 7
15:           $\theta \leftarrow \theta - \alpha \nabla_\theta L(\theta)$
16:         **end for**
17:     **end if**
18:     **if** update-interval steps have passed **then**
19:         $\theta_i^- \leftarrow \theta, \forall i = 1, 2.., m$
20:     **end if**
21: **end for**

---

Algorithm 1 presents the EMIX framework (in green) combined with QMIX Rashid et al. (2018), an off-the-shelf MARL algorithm. The total $Q$-value $Q_{tot}^\theta$ is computed by the mixer network with its inputs as the $Q$-values of all the agents conditioned on $s$ via the hypernetworks. Similarly, the target mixers approximate $Q_i^{\theta^-}$ conditioned on $s'$. In order to evaluate surprise within agents, we compute the standard deviations $\sigma$ and $\sigma'$ across all observations $z$ and $z'$ for each agent using $s$ and $s'$ respectively. The surprise value function, called the Surprise-Mixer, estimates surprise $V_{\text{surp}}^a(s, u, \sigma)$ conditioned on $s$, $u$ and $\sigma$. The same computation is repeated using the Target-Surprise-Mixer for estimating surprise $V_{\text{surp}}^a(s', u', \sigma')$ within next-states in the batch. Application of the energy operator along the non-singleton agent dimension for $V_{\text{surp}}^a(s, u, \sigma)$ and $V_{\text{surp}}^a(s', u', \sigma')$ yields the energy ratio $E$ which is used in Equation 7 to evaluate $L(\theta)$. We then use batch gradient descent to update parameters of the mixer $\theta$. Target parameters $\theta_i^-$ are updated every $update - interval$ steps.

## 5 EXPERIMENTS

Our experiments aim to evaluate the theoretical claims presented by EMIX along with its performance to prior MARL methods. Specifically, we aim to answer the following questions- (1) How does the provision of an EBM for surprise minimization compare to current MARL methods?, and (2) Does the algorithm validate the theoretical claims corresponding to its components?

### 5.1 ENERGY-BASED SURPRISE MINIMIZATION

We assess the validity of EMIX, when combined with QMIX, on multi-agent StarCraft II microman-agement scenarios (Samvelyan et al., 2019) as these consist of a larger number of agents with different action spaces. This in turn motivates a greater deal of coordination. Additionally, micromanagement scenarios in StarCraft II consist of multiple opponents which introduce a greater degree of surprise within consecutive states.

We compare our method to prior methods namely; (1) QMIX (Rashid et al., 2018), constituting of nonlinear value function factorization with monotonicity constraints; (2) Value Decomposition Networks (VDN) (Sunehag et al., 2018), consisting of linear additive factorization of $Q$ function; (3) Counterfactual Multi-Agent Policy Gradients (COMA) (Foerster et al., 2017), which consist of

| Scenarios | EMIX | SMiRL-QMIX | QMIX | VDN | COMA | IQL |
|---|---|---|---|---|---|---|
| 2s_vs_1sc | 90.33 ± 0.72 | 88.41 ± 1.31 | 89.19 ± 3.23 | 91.42 ± 1.23 | **96.90 ± 0.54** | 86.07 ± 0.98 |
| 2s3z | **95.40±0.45** | 94.93±0.32 | **95.30±1.28** | 92.03±2.08 | 43.33±2.70 | 55.74±6.84 |
| 3m | **94.90±0.39** | 93.94±0.22 | 93.43±0.20 | 94.58±0.58 | 84.75±7.93 | 94.79±0.50 |
| 3s_vs_3z | **99.58±0.07** | 97.63±1.08 | **99.43±0.20** | 97.90±0.58 | 0.21±0.54 | 92.32±2.83 |
| 3s_vs_4z | **97.22±0.73** | 0.24±0.11 | 96.01±3.93 | 94.29±2.13 | 0.00±0.00 | 59.75±12.22 |
| 3s_vs_5z | 52.91±11.80 | 0.00±0.00 | 43.44±7.09 | **68.51±5.60** | 0.00±0.00 | 18.14±2.34 |
| 3s5z | **88.88±1.07** | 88.53±1.03 | **88.49±2.32** | 63.58±3.99 | 0.25±0.11 | 7.05±3.52 |
| 8m | **94.47±1.38** | 89.96±1.42 | **94.30±2.90** | 90.26±1.12 | 92.82±0.53 | 83.53±1.62 |
| 8m_vs_9m | **71.03±2.69** | 69.90±1.94 | 68.28±2.30 | 58.81±4.68 | 4.17±0.58 | 28.48±22.38 |
| 10m_vs_11m | 75.35±2.30 | **77.85±2.02** | 70.36±2.87 | 71.81±6.50 | 4.55±0.73 | 32.27±25.68 |
| so_many_baneling | **95.87±0.16** | 93.61±0.94 | 93.35±0.78 | 92.26±1.06 | 91.65±2.26 | 74.97±6.52 |
| 5m_vs_6m | **37.07±2.42** | 33.27±2.79 | 34.42±2.63 | 35.63±3.32 | 0.52±0.13 | 14.78±2.72 |

Table 1: Comparison of success rate percentages between EMIX and prior MARL methods on StarCraft II micromanagement scenarios. EMIX is comparable to or improves over QMIX agent. In comparison to SMiRL-QMIX, EMIX demonstrates improved minimization of surprise. Results are averaged over 5 random seeds.

counterfactual actor-critic updates in a centralized critic; and (4) Independent $Q$ Learning (IQL) (Tan, 1993), wherein each agent acts independent of other agents. (5) In order to compare our surprise minimization scheme against pre-existing mechanisms, we compare EMIX additionally to a model-free implementation of SMiRL (Berseth et al., 2019) in QMIX. We use the generalized version of SMiRL as it demonstrates reduced variance across batches (Chen, 2020). This implementation is denoted as SMiRL-QMIX for comparisons. Details related to the implementation of EMIX are presented in Appendix D.

Table 1 presents the comparison of success rate percentages between EMIX and prior MARL algorithms on 12 StarCraft II micromanagement scenarios. Corresponding to each scenario, algorithms demonstrating higher success rate values in comparison to other methods have their entries highlighted in **bold** (see Appendix E.2 for a statistical analysis). Out of the 12 scenarios considered, EMIX presents higher success rates on 9 of these scenarios depicting the suitability of the proposed approach. In cases of *so_many_baneling* and *5m_vs_6m* having large number of opponents and a greater level of surprise, EMIX aptly improves over prior methods.

When compared to QMIX, EMIX depicts improved success rates on all of the 12 scenarios. On comparing EMIX with SMiRL-QMIX, we note that EMIX demonstrates a higher average success rate. This highlights the suitability of the energy-based scheme in the case of a larger number of agents and complex environment dynamics for surprise minimization.

## 5.2 ABLATION STUDY

We now present the ablation study for the various components of EMIX. Our experiments aim to determine the effectiveness of the energy-based surprise minimization method. Additionally, we also aim to evaluate the utility of dual approximators for surprise estimation in accordance with the precept from RL literature (Hasselt et al., 2016; Fujimoto et al., 2018; Haarnoja et al., 2018b).

### 5.2.1 EMIX OBJECTIVE

To weigh the effectiveness of energy-based scheme, we ablate the energy operator $\mathcal{T}$ and only utilize $V_{\text{surp}}^a$. Since this implementation employs dual approximators $V_{\text{surp},(i)}^a$ $i \in \{1, 2\}$ for stability, we call this implementation as TwinQMIX. Thus, we compare between QMIX, TwinQMIX and EMIX to assess the contributions of each of the proposed methods.

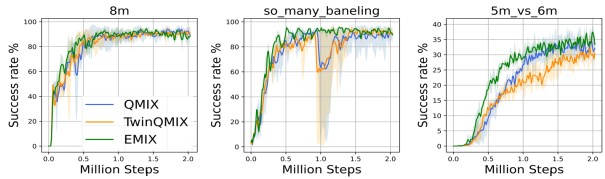

Figure 3: Ablations for each of EMIX's component. When compared to QMIX, EMIX and TwinQMIX depict improvements in performance and sample efficiency.

Figure 3 presents the comparison of average success rates for QMIX, TwinQMIX and EMIX on 3 different scenarios. Agents were evaluated for a total of 2 million timesteps with the lines in the plot indicating average success rates and the

shaded area as the deviation across 5 random seeds. In comparison to QMIX, TwinQMIX adds stability to the original objective by incorporating surprising estimates in the initial QMIX objective. On comparing TwinQMIX to EMIX we note that dual approximators play little role in improving convergence. Thus, the energy-based surprise minimization scheme is the main facet for significant performance improvement in the modified EMIX objective. This is demonstrated in the *5m_vs_6m* scenario wherein the EMIX implementation improves the performance of TwinQMIX in comparison to QMIX by utilizing a surprise-robust policy. In the case of *so_many_baneling* scenario which consists of a large number of opponents (27 banelings), EMIX tackles surprise effectively by preventing a significant drop in performance which is observed in cases of QMIX and TwinQMIX. We conjecture that this is a direct consequence of underestimations arising from $V^a_{\text{surp},(i)}$ estimates.

### 5.2.2 SURPRISE MINIMIZATION WITH TEMPERATURE

The importance of $\beta$ can be validated by assessing its usage in surprise minimization. However, it is difficult to evaluate surprise minimization directly as surprise value function estimates $V^a_{\text{surp}}(s, u, \sigma)$ vary from state-to-state across different agents and thus, they present high variance during agent's learning. We instead observe the variation of $E$ as it is a collection of surprise-based sample estimates across the batch. Additionally, $E$ consists of prior samples $V^a_{\text{surp}}(s, u, \sigma)$ for $V^a_{\text{surp}}(s', u', \sigma')$ which makes inference across different agents tractable.

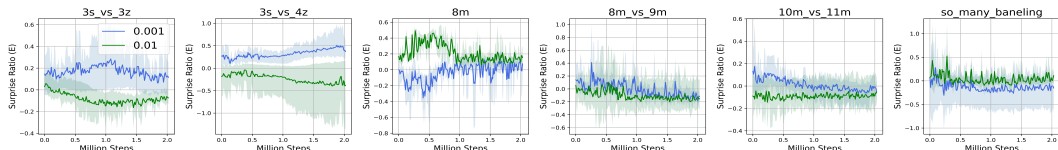

Figure 4: Variation of surprise minimization with temperature $\beta$. Learning of surprise is achieved by making use of a suitable value of temperature parameter ($\beta = 0.01$) which controls the stability in surprise minimization by utilizing $E$ as intrinsic motivation.

Figure 4 presents the variation of Energy ratio $E$ with the temperature parameter $\beta$ during learning. We compare two stable variations of E at $\beta = 0.001$ and $\beta = 0.01$. The objective minimizes $E$ over the course of learning and attains thermal equilibrium with minimum energy. Intuitively, equilibrium corresponds to convergence to optimal policy $\pi^*$ which validates the claim in Theorem 2. With $\beta = 0.01$, EMIX presents improved convergence and surprise minimization for 5 out of the 6 considered scenarios, hence validating the suitable choice of $\beta$. On the other hand, a lower value of $\beta = 0.001$ does little to minimize surprise across agents.

## 6 DISCUSSION

In this paper, we presented an energy-based perspective towards surprise minimization in multi-agent RL. Towards this goal we introduce EMIX, an energy-based intrinsic motivation framework for surprise minimization in MARL algorithms. EMIX utilizes a temporal EBM to estimate and minimize surprise jointly across all agents. Our theoretical claims on the formulation of minimization of temporal energy with surprise are corroborated upon utilizing EMIX on a suite of challenging MARL tasks requiring significant collaboration under fast-paced dynamics.

While EMIX serves as a practical example of EBMs in cooperative MARL, it presents several new avenues for future work. We shed light on 3 such aspects,

(1) Provision of an energy-based model naturally raises the question of *how can we efficiently sample from the surprise distribution?* Advances in sampling methods depict promise towards this aspect.

(2) Although suitable for lower dimensions, the scalability of EBMs towards high dimensional action spaces remains an open question. We conjecture that the utility of density-based methods and generative models can address the scalability gap.

(3) Lastly, the extension of an EBM framework to opponent-aware and competitive MARL settings presents a suitable tangent for learning multi-agent roles. Provision of an EBM for learning minimum energy role configurations would do away with the need for multi-stage training and complex exploration strategies. We leave the aforesaid as potential directions for future work.

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

## A    PROOFS

**Theorem 1.** *Given a surprise value function $V_{\text{surp}}^a(s, u, \sigma)\,\forall a \in N$, the energy operator $\mathcal{T}V_{\text{surp}}^a(s, u, \sigma) = \log \sum_{a=1}^{N} \exp\left(V_{\text{surp}}^a(s, u, \sigma)\right)$ forms a contraction on $V_{\text{surp}}^a(s, u, \sigma)$.*

*Proof.* We follow the process of (Asadi & Littman, 2017). Let us first define a norm on surprise values $||V_1 - V_2|| \equiv \max_{s,u,\sigma}|V_1(s, u, \sigma) - V_2(s, u, \sigma)|$. Suppose $\epsilon = ||V_1 - V_2||$,

$$\log \sum_{a=1}^{N} \exp\left(V_1(s, u, \sigma)\right) \leq \log \sum_{a=1}^{N} \exp\left(V_2(s, u, \sigma) + \epsilon\right)$$

$$= \log \sum_{a=1}^{N} \exp\left(V_1(s, u, \sigma)\right) \leq \log \exp\left(\epsilon\right) \sum_{a=1}^{N} \exp\left(V_2(s, u, \sigma)\right)$$

$$= \log \sum_{a=1}^{N} \exp\left(V_1(s, u, \sigma)\right) \leq \epsilon + \log \sum_{a=1}^{N} \exp\left(V_2(s, u, \sigma)\right)$$

$$= \log \sum_{a=1}^{N} \exp\left(V_1(s, u, \sigma)\right) - \log \sum_{a=1}^{N} \exp\left(V_2(s, u, \sigma)\right) \leq ||V_1 - V_2|| \tag{8}$$

Similarly, using $\epsilon$ with $\log \sum_{a=1}^{N} \exp\left(V_1(s, u, \sigma)\right)$,

$$\log \sum_{a=1}^{N} \exp\left(V_1(s, u, \sigma) + \epsilon\right) \geq \log \sum_{a=1}^{N} \exp\left(V_2(s, u, \sigma)\right)$$

$$= \log \exp\left(\epsilon\right) \sum_{a=1}^{N} \exp\left(V_1(s, u, \sigma)\right) \geq \log \sum_{a=1}^{N} \exp\left(V_2(s, u, \sigma)\right)$$

$$= \epsilon + \log \sum_{a=1}^{N} \exp\left(V_1(s, u, \sigma)\right) \geq \log \sum_{a=1}^{N} \exp\left(V_2(s, u, \sigma)\right)$$

$$= ||V_1 - V_2|| \geq \log \sum_{a=1}^{N} \exp\left(V_2(s, u, \sigma)\right) - \log \sum_{a=1}^{N} \exp\left(V_1(s, u, \sigma)\right) \tag{9}$$

Results in Equation 8 and Equation 9 prove that the energy operation is a contraction. $\square$

**Theorem 2.** *Upon agent's convergence to an optimal policy $\pi^*$, total energy of $\pi^*$, expressed by $E^*$ will reach a thermal equilibrium consisting of minimum surprise among consecutive states $s$ and $s'$.*

*Proof.* We begin by initializing a set of $M$ policies $\{\pi_1, \pi_2 ..., \pi_M\}$ having energy ratios $\{E_1, E_2 ..., E_M\}$. Consider a policy $\pi_1$ with surprise value function $V_1$. $E_1$ can then be expressed as

$$E_1 = \log \left[ \frac{\sum_{a=1}^{N} \exp\left(V_1^a(s', u', \sigma')\right)}{\sum_{a=1}^{N} \exp\left(V_1^a(s, u, \sigma)\right)} \right]$$

Invoking Assumption 2 for $s$ and $s'$, we can express $V_1^a(s', u', \sigma') = V_1^a(s, u, \sigma) + \zeta_1$ where $\zeta_1$ is a constant. Using this expression in $E_1$ we get,

$$E_1 = \log \left[ \frac{\sum_{a=1}^{N} \exp\left(V_1^a(s, u, \sigma) + \zeta_1\right)}{\sum_{a=1}^{N} \exp\left(V_1^a(s, u, \sigma)\right)} \right]$$

$$E_1 = \log \left[ \frac{\exp\left(\zeta_1\right)\sum_{a=1}^{N} \exp\left(V_1^a(s, u, \sigma)\right)}{\sum_{a=1}^{N} \exp\left(V_1^a(s, u, \sigma)\right)} \right]$$

$$E_1 = \zeta_1$$

Similarly, $E_2 = \zeta_2, E_3 = \zeta_3..., E_M = \zeta_M$. Thus, the energy residing in policy $\pi$ is proportional to the surprise between consecutive states $s$ and $s'$. Clearly, an optimal policy $\pi^*$ is the one with minimum surprise. Mathematically,

$$\pi^* \geq \pi_1, \pi_2..., \pi_M \implies \zeta^* \leq \zeta_1, \zeta_2..., \zeta_M$$
$$= \pi^* \geq \pi_1, \pi_2..., \pi_M \implies E^* \leq E_1, E_2..., E_M$$

Thus, proving that the optimal policy consists of minimum surprise at thermal equilibrium. $\qquad\square$

# B    RELATION TO MAXIMUM ENTROPY FRAMEWORK

## B.1    SIMILARITIES & DIFFERENCES

We conceptually compare EMIX to the maximum entropy framework.

**Similarities:** Both methods utilize an auxilary objective as intrinsic motivation to tackle uncertainty. While the maximum entropy formulation assigns low energy to uncertain actions, our method assigns low energy to uncertain encoded representations od states (as presented in Fig. 2).

**Differences:** Our method differs from maximum entropy in its optimization process and learning scheme. The maximum entropy formulation aims to maximize entropy in the value function space so as to motivate exploration. Our proposed scheme, on the other hand, aims to minimize surprise in the low-dimensional representation space to obtain dynamics-aware robust policies.

## B.2    CONNECTION TO SOFT Q-LEARNING

The Soft Q-Learning objective with $V_{\text{soft}}^{\theta^-}(s')$ and $Q_{\text{soft}}(u, s; \theta)$ as state and action value functions respectively is given by-

$$J_Q(\theta) = \mathbb{E}_{s,u\sim R}\left[\frac{1}{2}\left(r + \gamma\mathbb{E}_{s'\sim R}[V_{\text{soft}}^{\theta^-}(s')] - Q_{\text{soft}}(u, s; \theta)\right)^2\right]$$

$$= J_Q(\theta) = \mathbb{E}_{s,u\sim R}\left[\frac{1}{2}\left(r + \gamma\mathbb{E}_{s'\sim R}\left[\log\sum_{u\in A}\exp Q_{\text{soft}}(u', s'; \theta^-)\right] - Q_{\text{soft}}(u, s; \theta)\right)^2\right]$$

The gradient of this objective can be expressed as-

$$\nabla_\theta J_Q(\theta) = \mathbb{E}_{s,u\sim R}\left[\left(r + \gamma\mathbb{E}_{s'\sim R}\left[\log\sum_{u\in A}\exp Q(u', s'; \theta^-)\right] - Q_{\text{soft}}(u, s; \theta)\right)\right]\nabla_\theta Q_{\text{soft}}(u, s; \theta)$$
(10)

And the gradient of the EMIX objective is obtained as-

$$L(\theta) = \mathbb{E}_{s,u,s'\sim R}\left[\frac{1}{2}\left(r + \gamma\max_{u'}Q(u', s'; \theta^-) + \beta\log\left(\frac{\sum_{a=1}^N \exp\left(V_{\text{surp}}^a(s', u', \sigma')\right)}{\sum_{a=1}^N \exp\left(V_{\text{surp}}^a(s, u, \sigma)\right)}\right) - Q(u, s; \theta)\right)^2\right]$$

$$\nabla_\theta L(\theta) = \mathbb{E}_{s,u,s'\sim R}\left[\left(r + \gamma\max_{u'}Q(u', s'; \theta^-)\right.\right.$$
$$\left.\left. + \beta\log\left(\frac{\sum_{a=1}^N \exp\left(V_{\text{surp}}^a(s', u', \sigma')\right)}{\sum_{a=1}^N \exp\left(V_{\text{surp}}^a(s, u, \sigma)\right)}\right) - Q(u, s; \theta)\right)\right]\nabla_\theta Q(u, s; \theta) \quad (11)$$

Comparing Equation 10 to Equation 11 we notice that Soft Q-Learning and EMIX are related to each other as they utilize EBMs. Soft Q-Learning makes use of a discounted energy function which downweights the energy values over longer horizons. Actions consisting of lower energy configurations are given preference by making use of $Q_{\text{soft}}(u, s; \theta)$ as the negative energy. On the other hand, EMIX makes use of a constant energy function weighed by $\beta$ which minimizes

surprise-based energy between consecutive states. Both the objectives can be thought of as energy minimizing models which search for an optimal energy configuration. Soft Q-Learning searches for an optimal configuration in the action space whereas EMIX favours optimal behavior on spurious states. In fact, EMIX can be realized as a special case of Soft Q-Learning if the mixer agent utilizes an energy-based policy and attains thermal equilibrium. This leads us to express Theorem 3.

**Theorem 3.** *Given an energy-based policy $\pi$ with its target function $V(s') = \log \sum_{u \in A} \exp Q(u', s'; \theta^-)$, the surprise minimization objective $L(\theta)$ reduces to the Soft Q-Learning objective $L(\theta_{\text{soft}})$ in the special case surprise absent between consecutive states, $\sum_{a=1}^{N} \exp (V_{\text{surp}}^a(s', u', \sigma')) = \sum_{a=1}^{N} \exp (V_{\text{surp}}^a(s, u, \sigma))$.*

*Proof.* We know that the EMIX objective is given by-

$$L(\theta) = \mathbb{E}_{s,u,s' \sim R} \left[ \frac{1}{2} \left( r + \gamma \max_{u'} Q(u'; s', \theta^-) + \beta \log \left( \frac{\sum_{a=1}^{N} \exp (V_{\text{surp}}^a(s', u', \sigma'))}{\sum_{a=1}^{N} \exp (V_{\text{surp}}^a(s, u, \sigma))} \right) - Q(u, s; \theta) \right)^2 \right]$$
(12)

Replacing the greedy policy term $\max_{u'} Q(u', s'; \theta^-)$ with the energy-based value function $V(s') = \log \sum_{u' \in A} \exp Q(u', s'; \theta^-)$, we get,

$$L(\theta) = \mathbb{E}_{s,u,s' \sim R} \left[ \frac{1}{2} \left( r + \gamma \mathbb{E}_{s' \sim R}[V(s')] + \beta \log \left( \frac{\sum_{a=1}^{N} \exp (V_{\text{surp}}^a(s', u', \sigma'))}{\sum_{a=1}^{N} \exp (V_{\text{surp}}^a(s, u, \sigma))} \right) - Q(u, s; \theta) \right)^2 \right]$$
(13)

$$= L(\theta) = \mathbb{E}_{s,u,s' \sim R} \left[ \frac{1}{2} \left( r + \gamma \mathbb{E}_{s' \sim R} \left[ \log \sum_{u' \in A} \exp Q(u', s'; \theta^-) \right] \right. \right.$$
$$\left. \left. + \beta \log \left( \frac{\sum_{a=1}^{N} \exp (V_{\text{surp}}^a(s', u', \sigma'))}{\sum_{a=1}^{N} \exp (V_{\text{surp}}^a(s, u, \sigma))} \right) - Q(u, s; \theta) \right)^2 \right]$$

At thermal equilibrium, $\sum_{a=1}^{N} \exp (V_{\text{surp}}^a(s, u, \sigma)) = \sum_{a=1}^{N} \exp (V_{\text{surp}}^a(s', u', \sigma'))$,

$$= L(\theta) = \mathbb{E}_{s,u,s' \sim R} \left[ \frac{1}{2} \left( r + \gamma \mathbb{E}_{s' \sim R} \left[ \log \sum_{u' \in A} \exp Q(u', s'; \theta^-) \right] \right. \right.$$
$$\left. \left. + \beta \log \left( \frac{\sum_{a=1}^{N} \exp (V_{\text{surp}}^a(s', u', \sigma'))}{\sum_{a=1}^{N} \exp (V_{\text{surp}}^a(s', u', \sigma'))} \right) - Q(u, s; \theta) \right)^2 \right]$$

$$= L(\theta) = \mathbb{E}_{s,u,s' \sim R} \left[ \frac{1}{2} \left( r + \gamma \mathbb{E}_{s' \sim R} \left[ \log \sum_{u' \in A} \exp Q(u', s'; \theta^-) \right] + \beta \log(1) - Q(u, s; \theta) \right)^2 \right]$$
(14)

$$= L(\theta) = \mathbb{E}_{s,u,s' \sim R} \left[ \frac{1}{2} \left( r + \gamma \mathbb{E}_{s' \sim R} \left[ \log \sum_{u' \in A} \exp Q(u', s'; \theta^-) \right] - Q(u, s; \theta) \right)^2 \right]$$
(15)

Equation 15 represents the Soft Q-Learning objective, hence proving the result. ☐

## C    CONVERGENCE ANALYSIS

We now analyze convergence of the surprise minimization scheme during policy optimization. For brevity, our notation denotes the modified Bellman operator as $\mathcal{B}$ obeying the standard assumptions of monotonicity and contraction (Bertsekas, 2018). Additionally, we consider the cumulative value $\hat{V}_k = r_k + G_k + \beta \log \sum_{a=1}^{N} \exp(V_{\text{surp},(k)}^a(s, u, \sigma))$ as the sum of state values $V_k = r_k + G_k$ and surprise energy values $\beta \log \sum_{a=1}^{N} \exp(V_{\text{surp},(k)}^a(s, u, \sigma))$ at $k^{\text{th}}$ Bellman update.

Consider $\left\|V_k - V^*\right\|_2$ with $V^*$ being the optimal value at convergence,

$$\left\|\hat{V}_k - V^*\right\|_2 \leq \left\|\mathcal{B}\hat{V}_{k-1} + \beta \log \sum_{a=1}^{N} \exp(V_{\mathrm{surp},(k)}^a) - V^*\right\|_2 \tag{16}$$

$$\leq \left\|\mathcal{B}^2\hat{V}_{k-2} + \beta \log \sum_{a=1}^{N} \exp(V_{\mathrm{surp},(k-1)}^a) + \beta \log \sum_{a=1}^{N} \exp(V_{\mathrm{surp},(k)}^a) - V^*\right\|_2 \tag{17}$$

$$\leq \left\|\mathcal{B}^2\hat{V}_{k-2} + \beta \left(\log \sum_{a=1}^{N} \exp(V_{\mathrm{surp},(k-1)}^a) + \log \sum_{a=1}^{N} \exp(V_{\mathrm{surp},(k)}^a)\right) - V^*\right\|_2 \tag{18}$$

$$\leq \left\|\mathcal{B}^2\hat{V}_{k-2} + \beta \left(\log \left[\sum_{a=1}^{N} \exp(V_{\mathrm{surp},(k-1)}^a)\right]\left[\sum_{a=1}^{N} \exp(V_{\mathrm{surp},(k)}^a)\right]\right) - V^*\right\|_2 \tag{19}$$

Thus, for $k$ iterations, we have,

$$\leq \left\|\mathcal{B}^k V_0 + \beta \left(\log \prod_{i=1}^{k} \left[\sum_{a=1}^{N} \exp(V_{\mathrm{surp},(i)}^a)\right]\right) - V^*\right\|_2 \tag{20}$$

$$= \left\|\mathcal{B}^k V_0 + \beta \left(\log \sum_{a=1}^{N} \left[\prod_{i=1}^{k} \exp(V_{\mathrm{surp},(i)}^a)\right]\right) - V^*\right\|_2 \tag{21}$$

$$= \left\|\mathcal{B}^k V_0 + \beta \left(\log \sum_{a=1}^{N} \left[\exp(\sum_{i=1}^{k} V_{\mathrm{surp},(i)}^a)\right]\right) - V^*\right\|_2 \tag{22}$$

We now absorb the sum of surprise values from time index $i = 1, .., k$ in a single variable $V_{\mathrm{tot}}^a$. Thus, using $V_{\mathrm{tot}}^a = \sum_{i=1}^{k} V_{\mathrm{surp},(i)}^a$ and utilizing the Triangle Inequality, we get,

$$= \left\|\mathcal{B}^k V_0 - V^*\right\|_2 + \left\|\beta \left(\log \sum_{a=1}^{N} [\exp(V_{\mathrm{tot}}^a)]\right)\right\|_2 \tag{23}$$

We now bound the two terms separately. Considering the first term and following the results of value iteration convergence (Bertsekas & Tsitsiklis, 1995),

$$\left\|\mathcal{B}^k V - V^*\right\|_2 \leq \gamma^k \left\|V - V^*\right\|_2 \tag{24}$$

$$= \gamma^k \left\|V + V_\mu - V_\mu - V^*\right\|_2 \tag{25}$$

wherein $V_\mu$ denotes an approximation to $V$. Utilizing the triangle inequality yields,

$$\leq \gamma^k \left\|V - V_\mu\right\|_2 + \gamma^k \left\|V_\mu - V^*\right\|_2 \tag{26}$$

The two terms are bounded using the convergence result of (Bertsekas, 2018).

$$= \gamma^k \sqrt{r_{\max}} + \gamma^k \sqrt{\frac{r_{\max}|\mathcal{S}|}{1 - \gamma}} \tag{27}$$

Now, considering the second term in Equation 23,

$$\leq \beta \left\|\log \sum_{a=1}^{N} \exp(V_{\mathrm{tot}}^a)\right\|_2 \tag{28}$$

$$= \beta \left\|\log \sum_{a=1}^{N} \exp(V_{\mathrm{tot}}^a) - \log \sum_{a=1}^{N} \exp(V_{\mathrm{tot}}^*) + \log \sum_{a=1}^{N} \exp(V_{\mathrm{tot}}^*)\right\|_2 \tag{29}$$

using the triangle inequality,

$$\leq \beta \left\| \log \sum_{a=1}^{N} \exp(V_{\text{tot}}^a) - \log \sum_{a=1}^{N} \exp(V_{\text{tot}}^*) \right\|_2 + \beta \left\| \log \sum_{a=1}^{N} \exp(V_{\text{tot}}^*) \right\|_2 \tag{30}$$

Since $\mathcal{T} = \log \sum_{a=1}^{N} \exp(V_{\text{tot}}^a)$ is a contraction following Theorem 1, for the first term we have,

$$\leq \beta\gamma \left\| V_{\text{tot}}^a - V_{\text{tot}}^* \right\|_2 + \beta \left\| \log \sum_{a=1}^{N} \exp(V_{\text{tot}}^*) \right\|_2 \tag{31}$$

The second term in the above relation is bounded due to the completeness assumption, $\left\| \log \sum_{a=1}^{N} \exp(V_{\text{tot}}^*) \right\|_2$.

$$\leq \beta\gamma \left\| V_{\text{tot}}^a - V_{\text{tot}}^* \right\|_2 + \beta\zeta \ , \ \zeta > 0 \tag{32}$$

Finally, combining Equation 27 and Equation 32 in Equation 23, we obtain the desired convergence bound.

$$\left\| V_k - V^* \right\|_2 \leq \gamma^k \left( \sqrt{r_{\max}} + \sqrt{\frac{r_{\max}|\mathcal{S}|}{1-\gamma}} \right) + \beta \left( \gamma \left\| V_{\text{tot}}^a - V_{\text{tot}}^* \right\|_2 + \zeta \right) \tag{33}$$

While the first term in Equation 33 denotes the convergence of policy optimization, the second term indicates the bounded convergence of surprise to ecological niches with finite (yet nonzero) surprising elements. The policy optimization process converges at a geometric rate $\mathcal{O}(\gamma^k)$ towards its stable fixed points. The surprise minimization process, on the other hand, demonstrates an annealing behavior which depends on the temperature parameter $\beta$. Furthermore, convergence to stable fixed point $V_{\text{tot}}^a$ is bounded in respect to each agents individual surprise values $V_{\text{tot}}^a$. This insight indicates that different agents converge towards different locally optimal values of surprise. Finally, the presence of constant $\zeta$ corroborates prior claims (Schwartenbeck et al., 2013; Friston, 2010) that agents continue to experience surprise irrespective of their convergence to minimum energy niches.

To further develop intuition for this claim, consider the special case wherein $\left\| V_{\text{tot}}^a - V_{\text{tot}}^* \right\|_2 \to 0$. Irrespective of global convergence among all agents, a finite yet small $\zeta$ continues to contribute to the upper bound of $\left\| V_k - V^* \right\|_2$.

**Role of $\beta$:** We further discuss the role of $\beta$ which is of balancing the terms at successive iterations. While the first term geometrically decays with $\mathcal{O}(\gamma^k)$ rate, the second term approaches a finite constant $\beta\zeta$ as $V_{tot}^a \to V_{tot}^*$. Irrespective of our choice of $\beta$, the LHS $\|V_k - V^*\|_2$ is upper bounded by a constant which validates the claims of minimum yet finite surprise values. We do note that a small $\beta$ is still desirable to remove any approximation errors in order to push $V_k \to V^*$. However, this comes at the cost of increased surprise if $\beta$ is not selected appropriately.

# D  IMPLEMENTATION DETAILS

## D.1  MODEL SPECIFICATIONS

**Architecture:** This section highlights model architecture for the surprise value function. At the lower level, the architecture consists of 3 independent networks called *state_net*, *q_net* and *surp_net*. Each of these networks consist of a single layer of 256 units with ReLU non-linearity as activations. Similar to the mixer-network, we use the ReLU non-linearity in order to provide monotonicity constraints across agents. Using a modular architecture in combination with independent networks leads to a richer extraction of joint latent transition space. Outputs from each of the networks are concatenated and are provided as input to the *main_net* consisting of 256 units with ReLU activations. The *main_net* yields a single output as the surprise value $V_{\text{surp}}^a(s, u, \sigma)$ which is reduced along the agent dimension by the energy operator. Alternatively, deeper versions of networks can be used in order to make the extracted embeddings increasingly expressive. However, increasing the number of layers does little in comparison to additional computational expense.

**Computation of $\sigma$:** The deviation $\sigma$ corresponds to the standard deviation across each dimension of the state $s$. Considering the state as a tensor of size $B \times A \times M$ with $B$ as the batch size, $A$ as the number of agents and $M$ as the observation dimension, we compute $\sigma$ by calculating the standard deviation across the $M$ dimension. This yields $\sigma$ as a $B \times A \times 1$ dimensional array.

**Computation of surprise estimates:** $V_{surp}$ denotes the surprise value function which quantifies the amount of surprise experienced by agents. Analogous to a $Q$ value function which provides estimates of returns, $V_{surp}$ provides estimate of surprise. Our framework learns $V_{surp}$ much like any other value function (using a neural network), but by additionally undergoing a $\log \sum \exp$ transformation to obey the fixed point property. This is achieved by realizing log-sum-exp as an energy operator $\mathcal{T} = \log \sum \exp$ which can be computed using standard computation libraries. Since our code is implemented in PyTorch, we implement this as `T_V = torch.logsumexp(V_surp, dim=1)`.

## D.2    HYPERPARAMETERS

Table 2 presents hyperparameter values for EMIX. A total of 2 target $Q$-functions were used as the model is found to be robust to any greater values.

| Hyperparameters | Values |
|---|---|
| batch size | $b = 32$ |
| learning rate | $\alpha = 0.0005$ |
| discount factor | $\gamma = 0.99$ |
| target update interval | 200 episodes |
| gradient clipping | 10 |
| exploration schedule | 1.0 to 0.01 over 50000 steps |
| mixer embedding size | 32 |
| agent hidden size | 64 |
| temperature | $\beta = 0.01$ |
| target $Q$-functions | 2 |

Table 2: Hyperparameter values for EMIX agents

## D.3    SELECTION & TUNING OF $\beta$

One can manually tune $\beta$ using a fine-grained hyperparameter search. We tune $\beta$ between 0.001 and 1 in intervals of 0.01 with best performance observed at $\beta = 0.01$. However, we find two additional methods helpful for obtaining more accurate values. These are described as follows-

**Armijo's Line Search:** One can borrow from optimization theory and utilize Armijo's line search Nocedal & Wright (2006) by setting a termination condition. The method starts with a constant value of $\beta$ which is iteratively incremented/decremented until a termination criterion (example- $\|\nabla L(\theta)\| < \epsilon$ with $\epsilon$ a constant) is reached. While line search is proven to converge towards globally optimal values, its $\mathcal{O}(n^2)$ convergence may be computationally expensive that too in the MARL setting. Thus, we turn to the more efficient automatic tuning.

---

**Algorithm 2** Armijo's Line Search

1: Initialize $\beta, \delta \in (0, 1]$, EMIX & $\mathcal{T}V_{\text{surp}}^a$;
2: **while** EMIX$(Q + \beta * \mathcal{T}V_{\text{surp}}^a) >$ EMIX$(Q)$
   $+ \alpha * \beta * \nabla$EMIX$(\mathcal{Q})^{\mathrm{T}}\mathcal{T}V_{\text{surp}}^a$ **do**
3:     $\beta = \delta * \beta$
4: **end while**
5: return $\beta$

---

**Algorithm 3** Automatic Tuning

1: Initialize $\beta, \delta \in (0, 1]$, EMIX & $\mathcal{T}V_{\text{surp}}^a$;
2: EMIX$(Q + \beta * \mathcal{T}V_{\text{surp}}^a)$
3: beta_loss = $\beta * 0.5 * (\mathcal{T}V_{\text{surp}}^a - 0)^2$
4: beta_loss.backward()
5: return $\beta$

---

**Automatic Tuning:** We choose to automatically tune $\beta$ following single-agent RL literature Haarnoja et al. (2018b); Kumar et al. (2020). This is achieved by treating $\beta$ as a parameter and adaptively

optimizing over it using Adam. We treat a surprise value of $0$ as our target value. The method works well in practice and provides $\beta$ values closer to $0.01$ (our manual selection).

# E ADDITIONAL RESULTS

## E.1 QUALITATIVE ANALYSIS

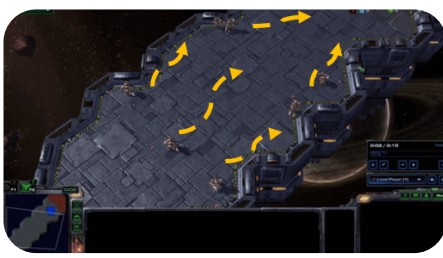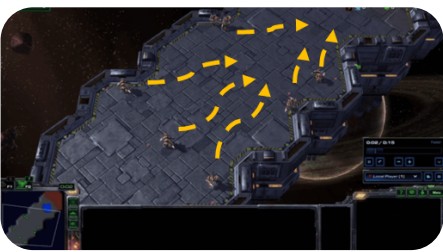

Figure 5: Task- *so_many_baneling*, **(left)** Behaviors learned by EMIX agents, **(right)** Behaviors learned by QMIX agents

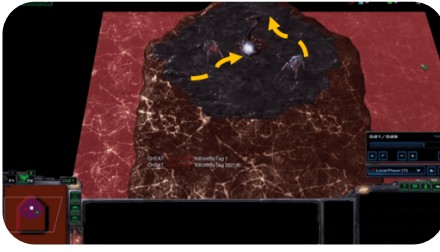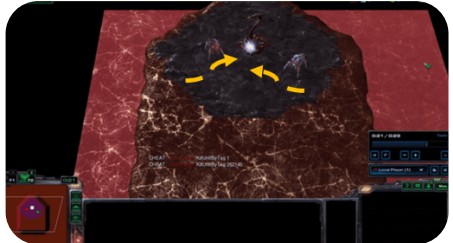

Figure 6: Task- *2s_vs_1sc*, **(left)** Behaviors learned by EMIX agents, **(right)** Behaviors learned by QMIX agents

We visualize and compare behaviors learned by surprise minimizing agents to the prior method of QMIX. Fig. 5 presents the comparison of EMIX and QMIX agent trajectories (in yellow arrows) on the challenging *so_many_baneling* task. The task consists of 27 baneling opponents which rapidly attack the agent team on a bridge. QMIX agents naively move to the central alley of the bridge and start attacking enemies early on. While QMIX agents naively maximize returns, EMIX agents learn a different strategy. EMIX agents rearrange themselves first at the corners of the bridge. Note that these corners provide cover from enemy's fire. Thus, EMIX agents learn to take cover before approaching the enemy head-on. This indicates that the surprise-robust policy is aware of the incoming fast-paced assault.

As another example, Fig. 6 presents behaviors on the *2s_vs_1sc* task wherein two agents must collaborate together to defeat a SpineCrawler enemy. The enemy, having a long tentacle pointing to the front, chooses to attack any one of the agents **randomly** in front of it. Additionally, the tentacle has a fixed length and cannot extend beyond this range. Random intermittent attacks indicate that the agents face a greater degree of surprise with no prior knowledge of the enemy's movement. We observe that QMIX agents take turns to attack the enemy by moving back and forth to minimize damage. EMIX agents, on the other hand, learn a different strategy. One of the EMIX agents stands at a distance to attack th enemy while the other agent goes around to attack from behind. This indicates that the policy is aware of enemy's limited movement.

## E.2 STATISTICAL SIGNIFICANCE

We follow the recommendation of Lones (2021) and evaluate the statistical significance of our results by carrying out the Mann-Whitney U test Mann & Whitney (1947). All 5 seeds of an algorithm

| Scenarios | EMIX | SMiRL-QMIX | QMIX | VDN | COMA | IQL |
|---|---|---|---|---|---|---|
| 2s_vs_1sc | 14 | 7 | - | 21 | **25** | 4 |
| 2s3z | **15** | 9 | - | 6 | 0 | 0 |
| 3m | **17** | 0 | - | 0 | 2 | 12 |
| 3s_vs_3z | **11** | 3 | - | 0 | 0 | 1 |
| 3s_vs_4z | **21** | 0 | - | 2 | 0 | 0 |
| 3s_vs_5z | 5 | 0 | - | **25** | 0 | 0 |
| 3s5z | 7 | **13** | - | 0 | 0 | 0 |
| 8m | **15** | 1 | - | 1 | 3 | 0 |
| 8m_vs_9m | 7 | **11** | - | 0 | 0 | 0 |
| 10m_vs_11m | 14 | **25** | - | 6 | 0 | 0 |
| so_many_baneling | **24** | 14 | - | 9 | 4 | 0 |
| 5m_vs_6m | **21** | 15 | - | 18 | 0 | 0 |

Table 3: Comparison of the $\mathcal{U}$ statistic on StarCraft II benchmark. $\mathcal{U}$ here denotes the statistical significance of an algorithm against QMIX (higher is better).

(on each task) are compared to that of QMIX to yield the $\mathcal{U}$ statistic. $\mathcal{U}$ here denotes the statistical significance of performance with higher values being desirable.

Table 3 presents the comparison of $\mathcal{U}$ statistic on the StartCraft II benchmark. EMIX demonstrates consistently high values of $\mathcal{U}$ across a diverse set of tasks when compared to SMiRL and prior MARL agents. This highlights the consistent surprise-minimizing performance of EMIX across random seeds.

### E.3 Additional Tasks

This section compares EMIX and TwinQMIX to prior MARL methods on the Predator-Prey tasks. In addition to the difficulty of task, we vary the number of opponents. This helps quantify the variation in performance against increasing level of surprise under fixed dynamics. Table 4 presents average returns. While all agents present comparable performance on the easier tasks, EMIX improves over QMIX and TwinQMIX on the more challenging *punish* and *hard* tasks. In the case of *punish*, EMIX is the only method to achieve greater than 20 returns outperforming baselines by a significant margin.

| Scenarios | EMIX | TwinQMIX | SMiRL-QMIX | QMIX | VDN | COMA | IQL |
|---|---|---|---|---|---|---|---|
| predator_prey_easy | **40.00 ± 0.13** | 40.00 ± 0.34 | 40.00 ± 0.98 | 40.00 ± 0.22 | 38.74 ± 0.64 | 27.49 ± 4.26 | 34.73 ± 2.92 |
| predator_prey | 40.00 ± 0.72 | 40.00 ± 1.92 | 40.00 ± 0.27 | **40.00 ± 0.16** | 36.23 ± 3.19 | 25.13 ± 0.92 | 31.59 ± 0.74 |
| predator_prey_punish | **24.17 ± 3.29** | 20.32 ± 4.15 | 19.31 ± 1.12 | 14.33 ± 3.81 | 17.21 ± 2.31 | 10.92 ± 4.35 | 7.86 ± 3.21 |
| predator_prey_hard | **12.34 ± 3.11** | 10.19 ± 1.15 | 10.47 ± 0.83 | 8.76 ± 4.33 | 5.19 ± 3.97 | -4.37 ± 1.53 | -9.26 ± 4.84 |

Table 4: Comparison of average returns between EMIX, its ablations and prior MARL methods on Predator-Prey tasks. EMIX improves over QMIX agent. In comparison to SMiRL-QMIX, EMIX demonstrates improved minimization of surprise. Results are averaged over 5 random seeds.

We consider a simple toy task from the Predator-Prey benchmark to demonstrate the importance of surprise minimization. We select *predator_prey_easy* due to its simplicity and convenient dynamics. The task consists of 3 agents and 3 opponents. We increase the number of opponents while keeping the task fixed. This way the dynamics of the MDP remain unchanged and the only changing factor is opponent behaviors.

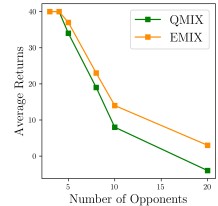

Fig. 7 presents the variation of average returns for EMIX and QMIX over 5 random seeds. While QMIX agents undergo a steady decrease in performance, EMIX agents are found robust to this fast degradation. Even after the addition of 20 opponents (against only 3 agents), EMIX is able to retain positive returns. The algorithm acquires a surprise robust-policy early on during training to tackle fast-paced changes introduced by the large number of agents.

Figure 7: Variation in performance with increasing number of agents.

### E.4 ADDITIONAL ABLATIONS

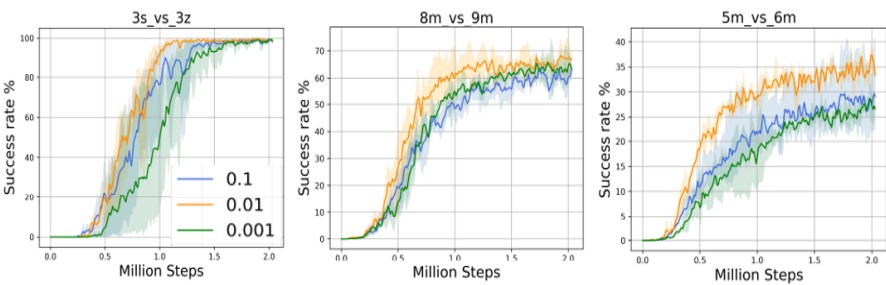

Figure 8: Variation in success rates with temperature $\beta$. A value of $\beta = 0.01$ is found to work best.

