# OpenReview forum: "Surprise Minimizing Multi-Agent Learning with Energy-based Models"
_ICLR.cc/2022/Conference — ICLR 2022 Submitted_

### Official Review · Reviewer_8Wiu · 2021-10-31

**Correctness:** 3
**Technical Novelty And Significance:** 2
**Empirical Novelty And Significance:** 2
**Recommendation:** 6
**Confidence:** 4

**Details Of Ethics Concerns:**

No.

**Main Review:**

## Strengths

1. The writing of this paper is clear enough and the presentation manner is concise.
2. The proposed method is novel to some extent.
3. The comparison between the proposed method and other related works is good.
4. Ablation studies are conducted to demonstrate the effects of hyperparameters.

## Weaknesses

1. Although the authors claimed that the proposed learning framework is for MARL, the whole theoretical framework is weakly connected to the context of MARL (in decentralized manner). The only connection is the assumption that the sum of suprising values instead of the single global surprising value (that can be seen as a centralized optimization that is equivalent to the single agent case). In my view, it could be more appropriate to write this paper as a single agent RL work and the contributions would not be reduced. For now, this work is actually a theory based on the single agent scenario and an trivial extension to the multi-agent scenario given the assumption of the global suprising value being equal to the sum of decentralized surprising values.
2. As the authors said in Theorem 3, the proposed surprising value objective can be reduced to the soft Q-learning objective. Could the authors explain in more details why the the proposed surprising value objective is necessary? The biggest concern from me is that it still needs to learn the thermal equilibrium of the surprising value. In other words, during the learning process the proposed surprising value is **not** constant, which may conflict with the claims in the paper.
3. In the concergence analysis, the authors subtly divided the suprising terms and the orginal value terms so that these two terms can be proved separately. **Nevertheless**, there exists a potential issue that the time scales of these two convergences (maybe inaccurate) are actually inconsistent. Specifically, the oprator of the orginal values is with respect to the dynamic environment, so it is corresponding to the transition between different states. While the oprator of the surprising value is actually for the static scenario (if I understand correctly), i.e., for a given state-action pair. Therefore, it could lead to a potential issue that the rate $\beta$ for the operator of surprising value may impact the convergence of optimal policies. Could the authors discuss more about this issue? The convergence rate of the whole sequence needs to be discussed in my view, according to the $\beta$.
4. The authors have compared the proposed algorithm with MARL algorithms, but most of them are irrelevant to the surprise term or intrinsic reward (as the authors claimed the relation in the paper). Can the authors show more comparisons with these highly relevant baselines so as to show the effectiveness of the proposed method?
5. About the SMAC experiment, some of the experimental results shown in the paper are far lower than the existing results, e.g. 5m\_vs\_6m, 3s\_vs\_5z, 10m\_vs\_11m. Can the authors give a convinciable reason or rerun these results by the latest baseline implementation from Pymarl.
6. As I concerned in theoretical analysis above, the choice of $\beta$ is important that is demonstrated in the abalation study. Can the authors show some laws or insights on how to select $\beta$?

**Summary Of The Paper:**

This paper introduced a suprise term in optimizing the policies (action-value functions in Q-learning) for solving the non-stationary challenge due to the rapid changes from the environment for MARL scenarios. This work not only proposed the concept of suprising value in the context of MARL, but also gave an operator (i.e. a contration) that can make the sequence of suprising values converges to a fixed point. The general sketch of the proofs is correct. Also, the authors showed that the convex conjugate of the suprising value operator is analogous to the minimization of the uncertainty among agents. By incorporating the suprising value into the Q-learning algorithm, the term led by the ratio between target and behaviour suprising value (suprising ratio) can also be interpreted as an intrinsic reward. Moreover, the authors also compare the proposed surprising minimization objective with the Soft Q-learning objective.

**Summary Of The Review:**

In summary, this work proposes an interesting problem and give a thourough analysis on the proposed method. However, due to the multiple concerns on the theoretical and experimental results, at the moment I can only recommend reject (but marginally below the threshold).

---

> ### Author Response · Authors · 2021-11-13
> **Response to Reviewer 8Wiu**
>
> Respected Reviewer,
>
> Thank you for providing detailed feedback on the paper which is of utmost value to our work. We address your concerns (highlighted in **bold**) below and in the general response comment above.
>
> **Although the authors claimed that the proposed learning framework is for MARL, the whole theoretical framework is weakly connected to the context of MARL (in decentralized manner). The only connection is the assumption that the sum of suprising values instead of the single global surprising value (that can be seen as a centralized optimization that is equivalent to the single agent case).**
> Yes, we agree that the only connecting link between our theoretical framework and the multi-agent setting is that of surprise value estimates. However, we emphasize that this link is enough to (1) setup a scalable MARL objective, (2) prove its theoretical convergence and (3) validate insights empirically. We expand on these points-
>
> (1) The aggregated surprise estimate collects actions from each agent to learn forthcoming changes. During training, backpropagation allows us to communicate these changes to all agents simulatenously. Note that each agent receives a centralized learning signal corresponding to the sum $\sum_{a=1}^{N}\exp(V^{a}_{surp})$.
>
> (2) Our theoretical analysis further shows that agents utilize this centralized training signal to converge to constant surprise minima at a geometric $\mathcal{O}(\gamma^{k})$ rate. This further highlights that MARL methods can theoretically benefit from EBMs which train agents in the centralized learning decentralized control paradigm.
>
> (3) Finally, our analysis presents the effectiveness of learning surprise in centralized manner. However, we also note that this brings benefits in decentralized control where EMIX agents demonstrate improved success rates when deployed to act in the environment.
>
> In the special case of only one agent, the objective reduces to the single-agent setting having the single global surprise value. However, in the case of multiple agents, the proposed surprise value formulation provides a general framework which may be extended to other RL paradigms (model-based, offline RL, etc). We also note that the SMiRL paper [1] provides a thorough empirical evaluation of surprise in the single-agent setting. Thus, focussing our work solely in this direction would defeat the purpose of a new and scalable framework.
>
> **Could the authors explain in more details why the proposed surprising value objective is necessary? The biggest concern from me is that it still needs to learn the thermal equilibrium of the surprising value.**
> The surprise value formulation provides a general method to reach equilibrium by minimizing energy. This interpretation of free energy minimization is pivotal as each point on the energy landscape can be thought of as a value of surprise (c.f. Figure 1). Agents move together on this energy landscape to minimize energy, i.e.- reach minimum surprising values. While reaching thermal equilibrium, energy continues to decrease and becomes constant at last. Analogously, surprise corresponding to each state is continuously minimized and becomes constant at the equilibrium. Thus, although the thermal equilibrium is being learned, it is approximated efficiently using a free energy minimization scheme to steer agents towards constant values.
>
> One can validate this mathematically. Upon examining the second term in Eq. 33 (Appendix C), we see that surprise stabilizes to a constant value. When $V^{a}\_{tot} \approx V^{*}\_{tot}$ the second approaches the constant $\zeta$. This indicates guaranteed convergence towards equilibrium with finite values for surprise.
>
> **There exists a potential issue that the time scales of these two convergences (maybe inaccurate) are actually inconsistent. Specifically, the operator of the orginal values is with respect to the dynamic environment, so it is corresponding to the transition between different states. While the operator of the surprising value is actually for the static scenario (if I understand correctly), i.e., for a given state-action pair. Therefore, it could lead to a potential issue that the rate $\beta$ for the operator of surprising value may impact the convergence of optimal policies. Could the authors discuss more about this issue?**
> Thank you for bringing this concern to our notice. While splitting Eq. (23) into its two parts, we introduce the notation $V^{a}\_{tot} = \sum\_{i=1}^{k} V^{a}\_{surp,(i)}$ which absorbs the time index $(i)$ inside the second operator. Thus, although both parts are proved under the same iterative procedure ($i=1,..,k$), it makes it difficult for the reader to track this. We have made this step more explicit by adding additional explanation before Eq. (23). With that said, we follow the steps and reach Eq. (33) where $V^{a}\_{tot}$ contains the time index $k$ and the analysis falls in line with the first term.

---

> > ### Author Response · Authors · 2021-11-13
> > **Response to Reviewer 8Wiu (continued)**
> >
> > "The role of $\beta$ is that of balancing the terms at successive iterations. While the first term geometrically decays with $\mathcal{O}(\gamma^{k})$ rate, the second term approaches a finite constant $\beta \zeta$ as $V^{a}\_{tot} \rightarrow V^{\ast}\_{tot}$. Irrespective of our choice of $\beta$, the LHS $\| V\_{k} - V^{\ast} \| \_{2}$ is upper bounded by a constant which validates our claims of minimum yet finite surprise values. We do note that a small $\beta$ may still be desirable to remove any approximation errors in order to push $V\_{k} \rightarrow V^{\ast}$. However, this comes at the cost of increased surprise if $\beta$ is not selected appropriately." The above discussion is added to Appendix C with additional details on the choice of $\beta$ added as Appendix D.3 following your suggestions.
> >
> > **The authors have compared the proposed algorithm with MARL algorithms, but most of them are irrelevant to the surprise term or intrinsic reward.**
> > We compare the proposed scheme to the more relevant SMiRL baseline which is proven to work well for single-agent settings. Additional uncertainty minimization methods either make use of world models or offline datasets, both of which will give the baseline an unfair advantage. In the case of MARL, unfortunately surprise minimization is still an active area of research and, to the best of our knowledge, we are not aware of any surprise minimization schemes with value function regularization. We sincerely apologize if we missed something out and welcome suggestions.
> >
> > **Can the authors give a convincible reason or rerun these results by the latest baseline implementation from Pymarl.**
> > Thank you for pointing this out. Our experiments use the latest PyMARL version with all baselines rerun from scratch for 5 random seeds. Note that we do not drop any seed nor change any hyperparameters. Additionally, it is worth noting that *5m_vs_6m* task can be significantly challenging over multiple runs (see results from [2]). Nevertheless, we run 3 additional seeds for each of these methods which are as follows-
> >
> > |Sscenario|EMIX|SMiRL-QMIX|QMIX|VDN|COMA|IQL|
> > |:-------:|:--:|:--------:|:--:|:-:|:--:|:-:|
> > |5m_vs_6m|__39.32$\pm$0.98__|34.21$\pm$1.23|35.76$\pm$2.44|32.59$\pm$2.83|1.13$\pm$1.01|11.31$\pm$1.79|
> > |3s_vs_5z|56.34$\pm$4.91|0.13$\pm$0.65|47.37$\pm$4.55|__67.38$\pm$2.26__|0.00$\pm$0.14|16.54$\pm$3.38|
> > |10m_vs_11m|73.28$\pm$1.13|__76.89$\pm$2.16__|72.53$\pm$1.64|69.85$\pm$3.43|5.67$\pm$3.76|27.19$\pm$10.22|
> >
> > This demonstrates the consistency of surprise minimization over a total of 8 random seeds on the above tasks.
> >
> > **Can the authors show some laws or insights on how to select $\beta$ ?**
> > Yes, we follow this suggestion based on your comment of convergence analysis. In practice, we may utilize two methods to select and tune $\beta$. We provide detailed formulation and discussions in Appendix D.3. Concisely, these are described as follows-
> >
> > __(1) Armijo's Line Search-__ One can borrow from optimization theory and utilize Armijo's line search [3] with termination conditions. The method starts with a constant value of $\beta$ which is iteratively incremented/decremented until a termination criterion (example- $\|\nabla L(\theta) \| < \epsilon$ with $\epsilon$ a constant) is reached. While line search is proven to converge towards globally optimal values, its $\mathcal{O}(n^{2})$ convergence may be computationally expensive that too in the MARL setting. Thus, we turn to the more efficient automatic tuning.
> >
> > __(2) Automatic Tuning-__ We choose to automatically tune $\beta$ following single-agent RL literature [4,5]. This is achieved by treating $\beta$ as a parameter and adaptively optimizing over it using Adam. We treat a surprise value of $0$ as our target value. The method works well in practice and provides $\beta$ values closer to $0.01$ (our manual selection).
> >
> > Please let us know if we have been able to address your concerns.
> >
> > [1]. Berseth et. al., SMiRL: Surprise Minimizing Reinforcement Learning in Unstable Environments, ICLR 2021.
> > [2]. Rashid et. al., Weighted QMIX: Expanding Monotonic Value Function Factorisation for Deep Multi-Agent Reinforcement Learning, NeurIPS 2020.
> > [3]. Nocedal \& Wright, Numerical Optimization, 2006.
> > [4]. Haarnoja et. al., Soft Actor-Critic: Off-Policy Maximum Entropy Deep Reinforcement Learning with a Stochastic Actor, ICML 2018.
> > [5]. Kumar et. al., Conservative Q-Learning for Offline Reinforcement Learning, NeurIPS 2020.

---

> > > ### Comment · Reviewer_8Wiu · 2021-11-20
> > > **Thanks for the author's response**
> > >
> > > Thanks for the author's response. Most of my concerns have been addressed.

---

> > > > ### Author Response · Authors · 2021-11-20
> > > > **Thank you**
> > > >
> > > > Thank you for raising your score, we are happy to know that your concerns were addressed.

---

> ### Author Response · Authors · 2021-11-18
> **Discussion**
>
> Kindly let us know if our response below addressed your concerns. We will be happy to answer if there are additional issues/questions.

---

### Official Review · Reviewer_g9cM · 2021-10-31

**Correctness:** 3
**Technical Novelty And Significance:** 3
**Empirical Novelty And Significance:** 3
**Recommendation:** 5
**Confidence:** 4

**Main Review:**

Strengths:
--The presentation is clear and easy to follow.
--The empirical results and ablation studies are sufficient.


Weaknesses (Questions):

1- The ``deviation $\sigma$ within states for each agent $a$ ” is not clear to me. As the authors only defined observations $O$ for each agent in Sec 3.1, is  $\sigma$ the deviation within all agents’ observations?


2- in Eq. (7), should the second $Q(u,s;θ) $ read as $\hat{Q}(u,s;θ)$?

3- Eq.(3) is not clear to me. The surprise is also not defined yet. Why should Eq(3) represent a global surprise? How minimizing Eq(7) can minimise surprise?

4- Table 1 shows that EMIX improves the winning rate in most of the considered tasks. But it seems the improvement is marginal. This is also the case for the results in Figure 3.

5- Only using SC-II micromanagement is not convincing enough. It is better to include results on other different benchmarks.


**Summary Of The Paper:**

This paper focuses on the value factorization approach for solving multi-agent problems. It  considered the spurious surprise states in execution and proposed to address this problem through the proposed free-energy minimization framework.


**Summary Of The Review:**


This paper is attempting to extend the surprise minimisation to the multi-agent learning regime. My major concern is that Eq.(4) seems to be solely a marginally new way of representing the Q-function. It is not different from the typical Q-Mix or Q learning.

The core message of how Eq(4) or Eq(7) can minimise surprise is not clear to me either.

---

> ### Author Response · Authors · 2021-11-13
> **Response to Reviewer g9cM**
>
> Respected Reviewer,
>
> Thank you for providing detailed feedback on the paper which is of utmost value to our work. We address your concerns (highlighted in **bold**) below and in the general response comment above.
>
> **The "deviation $\sigma$ within states for each agent $a$" is not clear to me. As the authors only defined observations  for each agent in Sec 3.1, is  the deviation within all agents’ observations?**
> Apologies for the confusion. The deviation corresponds to the "standard deviation $\sigma$ of observations for each agent $a$". We have updated the sentence with this expression. Additionally, we have added a short note on teh computation of $\sigma$ in Appendix D.1-   "Considering the state as a tensor of size $B \times A \times M$ with $B$ as the batch size, $A$ as the number of agents and $M$ as the observation dimension, we compute $\sigma$ by calculating the standard deviation across the $M$ dimension. This yields $\sigma$ as a $B \times A \times 1$ dimensional array."
>
>
> **In Eq. (7), should the second $Q(u,s,;\theta)$ read as $\hat{Q}(u,s;\theta)$?**
> Following the equation prior to Eq. (7), we collect the $\log\sum_{a=1}^{N}\exp(V^{a}_{surp}(s,u;\sigma))$ in the new term $E$. This keeps the Q value in Eq. (7) as $Q(u,s;\theta)$. Since this may confuse the reader, we have added an extra intermediate step before Eq. (7) with a short description.
>
> **Eq.(3) is not clear to me. The surprise is also not defined yet. Why should Eq (3) represent a global surprise?**
> We approximate surprise using the surprise value function $V^{a}_{surp}(s,u,\sigma)$ (defined in para before Eq (3)). Eq (3) characterizes the energy operation defined as a logsumexp (commonly known as the soft-maximum) function. The intuition here is as follows-
>
> If we were to take a hard maximum ($\underset{a}{\max}$) then the function would be dominated by the most active agents as these would have high values of $V^{a}_{surp}(s,u,\sigma)$. Instead, we take a soft-maximum by summing across all agent's exponential surprise values. This allows the function to induce information from all agents in the multi-agent population. Thus, $\mathcal{T}$ acts as a global estimate of surprise in contrast to the local $\underset{a}{\max}$ estimate.
>
> **Only using SC-II micromanagement is not convincing enough. It is better to include results on other different benchmarks.**
> Thank you for the suggestion. We have conducted additional experiments on the Predator-Prey domain in Appendix E.3. These tasks range from simple to challenging scenarios consisting of varying number of opponents. Table 4 presents the returns for all methods averaged across 5 random seeds. EMIX continues to retain improvements over prior surprise minimization and general MARL methods. Note that in the challenging *punish* scenario with high negative reward penalties, the energy-based framework is the only algorithm to obtain > 20 consistent returns.
>
> **My major concern is that Eq.(4) seems to be solely a marginally new way of representing the Q-function. The core message of how Eq(4) or Eq(7) can minimise surprise is not clear to me either.**
> Eq. (7) presents a general and simple objective for minimizing surprise as it regularizes the value function by learning an additional degree of uncertainty (using $\sigma$) across evolving dynamics. Note that this is in opposition to single-agent methods (for example- RND [1], VIME [2] and hDQN [3]) which either require multiple architectures, estimation of non-trivial Information Theoretic quantities or hand-engineered subgoals respectively.
>
> The energy-based scheme only utilizes Eq. (3) to estimate an aggregate of surprise. This estimate is then used alongside $Q$ in Eq. (4) to train the agent. Since Eq. (4) utilizes Eq. (3) as intrinsic regularization, the algorithm simultaneously minimizes this learned uncertainty while maximizing returns. The complete objective is formulated in Eq. (7) where we collect the surprise minimization terms in variable $E$. $E$ here forms a normalized probability distribution over $V^{a}_{surp}(s,u,\sigma)$ which is learned over the course of training.
>
> Please let us know if we have been able to address your concerns.
>
> [1]. Burda et. al., Exploration by random network distillation, 2018.
> [2]. Houthooft et. al., VIME: Variational Information Maximizing Exploration, 2016.
> [3]. Kulkarni et. al., Hierarchical deep reinforcement learning: Integrating temporal abstraction and intrinsic motivation, NeurIPS 2016.

---

> > ### Comment · Reviewer_g9cM · 2021-11-18
> > **Followup questions**
> >
> > Thanks for the authors' clarifications. That helps clarify some of my questions.
> >
> > Some major concern that I hold with this paper are that.
> >
> > 1) the definition of surprise is not clear, thus not sensible for the reviewer to evaluate its soundness. The authors define that $V^a_{surp}(s,u,\sigma)$ defines the surprise, but what is "surprise"?
> >
> > 2) why minimising the loss function Eq(7) can minimise the surprise? It seems only learning a new $Q$-function $\hat{Q}$ and cannot guarantee that $V^a_{surp}(s,u,\sigma)$ is minimised;
> >
> > 3) I do not understand the usage of $\sigma$. In the considered Dec-POMDP, it is not surprising if $\sigma$ is large or small. What is the relation between the surprise and $\sigma$? How is $V^a_{surp}(s,u,\sigma)$ defined? The author mentioned "soft-maximum operator", but it is helpful to give the details in this paper as it is important.

---

> > > ### Author Response · Authors · 2021-11-18
> > > **Response to Followup questions**
> > >
> > > Thank you for the followup questions. We further clarify your concerns in detail-
> > >
> > > __the definition of surprise is not clear, thus not sensible for the reviewer to evaluate its soundness. The authors define that $V^{a}\_{surp}(s,u,\sigma)$ defines the surprise, but what is "surprise"?__
> > > We do provide an explicit and succinct definition of surprise in Abstract (line 3) "quantifies the degree of change in an agent’s environment". We have added additional discussion on this in Section 1 for reader's convenience. Additionally, we note that our definition of surprise is based on prior works in surprise minimization [1].
> > >
> > > $V^{a}\_{surp}(s,u,\sigma)$ denotes an estimation of surprise. Statistically, surprise may be quantified as any metric of uncertainty $\text{Unc(s,a)}$ within the agent's state-action distribution [2]. This metric quantifies the change in states and actions based on agent's exploration of the MDP. A good example of this is entropy of the policy $\mathcal{H}(\pi)$ [3] which denotes the degree of uncertainty in agent's action distribution. Another example is a world model $\mathbb{P}(s^{\prime}|s,a)$ [4] which denotes the change in agent's state distribution. Analogously, our setting considers the quantifier $\mathcal{T}V^{a}\_{surp}(s,u,\sigma) = \log\sum\exp(V^{a}\_{surp})$ which quantifies the changes in states, actions and standard deviations $\sigma$ of the observations.
> > >
> > > __why minimising the loss function Eq(7) can minimise the surprise? It seems only learning a new $Q$-function $\hat{Q}$ and cannot guarantee that is $V^{a}\_{surp}(s,u,\sigma)$ minimised__
> > > Thank you for following up on this. We consider both aspects of your concern separately-
> > >
> > > __Minimization of Surprise__- To understand Eq.(7), we must start with Eq. (3). Eq. (3) provides an aggregated estimate of surprise. We plug this estimate in Eq. (4) to train the agent. This way, the agent uses $\hat{Q}$ instead of $Q$. In the absence of EMIX, the agent would simply maximize $Q$ (expected returns). However, in the presence of surprise minimization the agent is constrained by the second regularization term $\log\sum\exp(V^{a}\_{surp})$. While maximizing $Q$, the agent *must also minimize $\log\sum\exp(V^{a}\_{surp})$* which acts as an aggregated penalty of surprise from all agents. This is exactly reflected in Eq. (7) which presents the TD learning objective of $\hat{Q}$ values. Eq. (7) is the objective which is learned using batch gradient descent.
> > >
> > > __Guaranteed Minimization__- Following our above intuitiion, we further study the guaranteed minimization of surprise. Our study provides a thorough convergence analysis (a lacking component in prior works) of the surprise minimization scheme in Appendix C. It is also worth noting this guarantee is general and holds for *any* MARL as well as single-agent setting. Upon analyzing Eq. (33), one readily notices that aggregated surprise converges to a constant ($\beta\zeta$) in the limit of infinite updates and a good approximation scheme. This corroborates with our theoretical claims that upon reaching an energy-equilibrium, surprise stagnates to minimum yet finite values. We emphasize that this guarantee acts as a certificate for minimization of $V^{a}\_{surp}$ to a local minimum niche.
> > >
> > > __I do not understand the usage of $\sigma$. In the considered Dec-POMDP, it is not surprising if $\sigma$ is large or small. What is the relation between the surprise and $\sigma$ ? How is $V^{a}\_{surp}(s,u,\sigma)$ defined? The author mentioned "soft-maximum operator", but it is helpful to give the details in this paper as it is important.__
> > > Thank you for the suggestion. We have added additional explanations on the definition of surprise (Section 1), the role of $\sigma$ (Section 4.1) and intuition behind $V^{a}\_{surp}$ (Section 4.1). "In the case of high-dimensional state spaces (such as multiple opponents), $\sigma$ informs agents of the abrupt statistical change that would take place upon executing action $u$. We formulate surprise as $\mathcal{T} V^{a}\_{surp}(s,u,\sigma)$ which serves as an uncertainty quantifier $\text{Unc(s,a)}$ of the state-action distribution. Here $V^{a}\_{surp}(s,u,\sigma)$ denotes the surprise value function which serves as a mapping from agent and environment dynamics to surprise."
> > >
> > >
> > > [1]. Berseth et. al., SMiRL: Surprise Minimizing Reinforcement Learning in Unstable Environments, ICLR 2021.
> > > [2]. Schwartenbeck et. al., Exploration, novelty,surprise, and free energy minimization, Frontiers in Pyschology, 2013.
> > > [3]. Haarnoja et. al., Soft actor-critic: Off-policy maximum entropy deep reinforcement learning with a stochastic actor, ICML 2018.
> > > [4]. Hafner et. al., Dream to Control: Learning Behaviors by Latent Imagination, ICLR 2020.

---

> > > ### Author Response · Authors · 2021-11-28
> > > **Discussion**
> > >
> > > Kindly let us know if our followup response below addressed your concerns.

---

### Official Review · Reviewer_zFND · 2021-11-01

**Correctness:** 2
**Technical Novelty And Significance:** 2
**Empirical Novelty And Significance:** Not applicable
**Recommendation:** 6
**Confidence:** 1

**Main Review:**

There is, what appears to be, a very interesting argument being made here and it draws together a number of strands of research from different disciplines. The multi-agent domain being a clear candidate where independent gradient ascent from individual agents can result in a myopic trap interfering with convergence to a global objective. Nevertheless, a number of the ideas, methods, equations and results in this paper are not as clear to me as I would like and so I cannot endorse the paper for acceptance.

One of the main themes that was running through the paper was the idea of surprise minimisation and how this has some degree of tension with the surprise maximisation aspect of maximum entropy (which as the authors say is a strategy for ensuring good exploration). However, this tension is never properly resolved to my satisfaction.

It is clear that the results show advantages over the other SoTA methods, but I am not sure that the method could be replicated by a reader and so I am unsure whether these experiments could be validated.

## Detailed notes

The term EBM needs defining when first used.

Should really cite the Dec-POMDP literature rather than the POMDP paper.

p2-3: Is  the joint action a cross product over identical action spaces? Or can each agent have different actions available.

If the observation function is a function (determined) then do all agents get the same observation at time t? Or is it probabilistic but the same distribution for all agents? Or does/can it depend on the agent?

The policy and value function seems to depend on the hidden state.

I think you mean that $\pi^\star_\theta = \pi_{\theta^\star}$ (notationall that the best policy. It's clear what was intended but the notation is sloppy.

In equation 1, b is not very clear. Is this the expectation of the sample average over the minibatch? In which case that is not what the equation says. In short, b should enter into the equation somewhere other than just the subscript to the expectation. Also, sampled from memory R is ill-defined.

Figure 1 does not clearly illustrate what it intends to. There is no description of what the grey surface is, nor of the blue points on this surface and the connecting lines. Yes, the little pictures of robots with a red background are associated with a region of lighter grey, and the green associated with darker grey (which we are told is associated with energy minima), but in all this doesn't leave the reader any the wiser as to what is going on.

One of the arguments to the surprise value function $V_{\text{surp}}$ is $\sigma$ which is described as "deviation...within states for each agent a", but it isn't clear what is meant by this.

The function $V_{\text{surp}}$ is not clear at all. It is said that the $\Tau$ operator is a log-sum-exp contraction for this function but it isn't clear how to calculate this in the first place. Maybe I am missing something.

**Summary Of The Paper:**

The authors present a method to regularise the learning of Q values within Decentralised partially observable markov decision processes, where the regulariser is one that minimises surprise in some way across the population of agents within the environment.

This it is argued allows the agents to avoid situations in which states are "rapidly changing", instead aiming to reach an equilibrium state where just enough surprise is being experienced as part of a reward maximising objective.

The authors present a series of results in which their method outperforms a number of SoTA alternatives on a reasonable looking set of benchmarks, as well as an ablation study showing the criticality of each proposed component.


**Summary Of The Review:**

I confess that I miss the point of the paper. I am not confident that I could recreate the method and this potentially of by-product of key bits of information being excluded from explanations. I am fairly confident that the authors have a strong technical understanding and justification for what they have done, but this is not expressed clearly in the paper, at least not in a way that I can decode.

## Update after rebuttal

My concerns have been partly addressed by clarification on what measures of surprise can be used and how these would be estimated. The publication of the code also goes some way to addressing my concerns. The other updates also improve the paper and the explanations and so I am changing my recommendation to weak accept.

---

> ### Author Response · Authors · 2021-11-13
> **Response to Reviewer zFND**
>
> Respected Reviewer,
>
> Thank you for providing detailed feedback on the paper which is of utmost value to our work. We address your concerns (highlighted in __bold__) below and in the general response comment above.
>
> **One of the main themes that was running through the paper was the idea of surprise minimisation. However, this tension is never properly resolved to my satisfaction.**
> We apologize for this confusion. We connect surprise minimization to two types of EBM formulations (in Section 2),
> (1) Standard Boltzmann formulations
> (2) Maximum entropy framework.
>
> The maximum entropy framework is indeed a special case of the Boltzmann formulation wherein the policy forms a Gibbs distribution over Q values, $\pi \propto \exp(Q)$. The framework consists of a reward and entropy maximization objective. This provisions effective exploration in the form of intrinsic motivation.  Note that the maximum entropy formulation does not link to surprise maximization since surprise is an extrinsic quantity obtained from interacting with environment dynamics. The maximum entropy framework only assigns high entropy in states where the policy is uncertain. To make things more clear, we have added the following discussion on similarities and differences from maximum entropy framework in Appendix B.1-
>
> **Similarities-** Both methods utilize an auxilary objective as intrinsic motivation to tackle uncertainty. While the maximum entropy framework assigns low energy to uncertain actions in the policy, our method assigns low energy to uncertain encoded representations of states (c.f. Figure 2).
>
> **Differences-** Algorithms differ in their optimization process and learning schemes. The maximum entropy formulation aims to maximize entropy in the value function space so as to motivate exploration. Our proposed scheme aims to minimize surprise in the low-dimensional representation space to obtain dynamics-aware robust policies.
>
> The paper additionally draws a connection to entropy minimization of conjugate (gradient) space in Remark 3. The main point of this discussion is to emphasize that minimizing surprise leads to minimizing uncertainty in the learning signal, hence producing a stable paradigm for end-to-end robut policy optimization.
>
> **I am not sure that the method could be replicated by a reader**
> We understand the concerns and firmly believe in reproducibility of our work. To this end, we will be open-sourcing our code implementation along with detailed notes on reproducibility. Further, we have made our code modular and easy-to-read along with a short blog post which would help the community better navigate and build upon it.
>
> **The term EBM needs defining when first used.**
> Yes, we define the Energy-based Model (EBM) in abstract and last paragraph of page 1. Additionally, a more formal introduction to EBMs is presented in Section 3.2.
>
> **Should really cite the Dec-POMDP literature rather than the POMDP paper.**
> Thank you for point this out, now fixed.
>
> **p2-3: Is the joint action a cross product over identical action spaces? Or can each agent have different actions available.**
> The joint action $u$ is a combination of different actions $u^{a}$ each corresponding to an action space $\mathcal{A}^{a}$ for each agent $a$. We now state this explicitly in Section 3.1.
>
> **If the observation function is a function (determined) then do all agents get the same observation at time t? Or is it probabilistic but the same distribution for all agents? Or does/can it depend on the agent?**
> The observation function $O(s,a)$ is determined and depends on the state $s$ and agent $a$. Thus, the function provides a unique observation to each agent $a$.
>
> **The policy and value function seems to depend on the hidden state.**
> The policy and value function are trained from raw state and action inputs as well as the latent representations extracted from EMIX encoders.
>
> **I think you mean that $\pi_{\theta}^{\ast}$ = $\pi_{\theta^\ast}$ (notationally that the best policy). It's clear what was intended but the notation is sloppy.**
> Our sincere apologies, the error has now been corrected.
>
> **In equation 1, b is not very clear. Is this the expectation of the sample average over the minibatch? In which case that is not what the equation says. In short, b should enter into the equation somewhere other than just the subscript to the expectation. Also, sampled from memory R is ill-defined.**
> We note that the notation $b$ can be confusing and switch to a simpler notation $\mathbb{E}_{s,u,s^{\prime}\sim \mathcal{R}}$. The subscript indicates that the state $s$, joint action $u$ and next-state $s^{\prime}$ are sampled from the replay buffer $\mathcal{R}$ at each update step.
> Eq. (1) and the replay buffer sentence are updated accordingly.

---

> > ### Author Response · Authors · 2021-11-13
> > **Response to Reviewer zFND (continued)**
> >
> > **One of the arguments to the surprise value function $V_{surp}$ is $\sigma$ which is described as "deviation...within states for each agent a", but it isn't clear what is meant by this.**
> > We rephrase the sentence to "standard deviation $\sigma$ of observations for each agent $a$". Additionally we have added the following details on computation of $\sigma$ in Appendix D.1-
> > "The deviation $\sigma$ corresponds to the standard deviation across each dimension of the state $s$. Considering the state as a tensor of size $B \times A \times M$ with $B$ as the batch size, $A$ as the number of agents and $M$ as the observation dimension, we compute $\sigma$ by calculating the standard deviation across the $M$ dimension. This yields $\sigma$ as a $B \times A \times 1$ dimensional array."
> >
> > **The function $V_{surp}$ is not clear at all. It is said that the $\mathcal{T}$ operator is a log-sum-exp contraction for this function but it isn't clear how to calculate this in the first place.**
> > "$V_{surp}$ denotes the surprise value function which quantifies the amount of surprise experienced by agents. Analogous to a $Q$ value function which provides estimates of returns, $V_{surp}$ provides estimate of surprise. Our framework learns $V_{surp}$ much like any other value function (using a neural network), but by additionally undergoing a $\log\sum\exp$ transformation to obey the fixed point property. This is achieved by realizing log-sum-exp as an energy operator $\mathcal{T} = \log\sum\exp$ which can be computed using standard computation libraries. Since our code is implemented in PyTorch, we implement this as `T_V = torch.logsumexp(V_surp, dim=1)`." The discussion has been further added to Appendix D.1.
> >
> > Please let us know if we have been able to address your concerns.

---

> > > ### Comment · Reviewer_zFND · 2021-11-29
> > > **Thank you**
> > >
> > > Thank you,
> > >
> > > This addresses most of my concerns. I think that my main barrier to acceptance was greater clarity on what could be used as surprise values and the repeatability of the experiments and these concerns have been addressed by the authors. Therefore I will change my recommendation to weak accept.

---

> > > > ### Author Response · Authors · 2021-11-29
> > > > **Thank you**
> > > >
> > > > Thank you! We are happy to know that your concerns were suitably addressed.

---

> ### Author Response · Authors · 2021-11-18
> **Discussion**
>
> Kindly let us know if our response below addressed your concerns. We will be happy to answer if there are additional issues/questions.

---

### Official Review · Reviewer_doix · 2021-11-03

**Correctness:** 4
**Technical Novelty And Significance:** 3
**Empirical Novelty And Significance:** 2
**Recommendation:** 6
**Confidence:** 3

**Main Review:**

This work builds in the interesting new directions around surprise minimization and energy based models.  Moreover, it provides a powerful theoretical framework incorporating ideas from energy based models and providing an algorithm with convergence grantees.  To the best of my knowledge the mathematical results seem correct, though I have not checked the work in the Appendix, they are intuitive and seem to be extensions of existing ideas.

Though the theoretical results are strong, the empirical results are not as compelling.  For an evaluation of this method, I would expect some sort of qualitative evaluation showing that the agents in fact behave in a way to minimize surprise.  The agents on their own should show the same qualitative behaviors seen in surprise minimization in single agent settings, but, more importantly, it is important to be clear about the qualitative differences we should expect in multi-agent settings and to verify that those effects occur since there is added complexity with the multi-agent aspects of the problem.

Without those qualitative effects, empirically the results seem to be marginal, for instance in Table 1 it is difficult to tell which of the results are statistically significant.  It seems like most of the error bars overlap, so the results are statistically insignificant, but only one of the methods is bolded.  To be clear, I do not think that the method needs to surpass all others on success rate, it isn't designed to make those sorts of improvements, but because that is all we have for empirical evaluation the best I can conclude is that the intervention into QMIX does not break the method.

On a related note, I think the work would be much more intuitive if the utility of minimizing surprise in multi-agent RL, and the unique challenges that come with multi-agent RL were made more clear from the beginning.  As it stands, I understand from the introduction I understand that EBM are a natural way to approach surprise minimization in multi-agent RL, but it isn't clear to me why you would want surprise minimization in multi-agent RL particularly, and what is different about multi-agent RL that would effect the sorts of behavior we would expect out of surprise minimization.  I think this could be alleviated by an example showing how minimizing surprise in MARL produces desirable effects, and that doing surprise minimization without considering any multi-agent aspects does not produce that desirable behavior.


Minor point:
* In the final paragraph of page 3, the Q_a is said to be lower bounded by Q^* when it is demonstrated to be upper bounded by Q^*

**Summary Of The Paper:**

This paper proposes a method in the setting of centralized training and decentralized execution for training policies which minimize surprise.  The method uses insights from energy based models, deriving an energy operator that forms a contraction operator on the appropriate value functions, and uses this to make a variant of QMIX, which promotes surprise minimization across the multi-agent system.

**Summary Of The Review:**

I will weakly recommend rejection due to the limited qualitative evaluation and otherwise inconclusive empirical results.  I would be willing to reconsider my score if the approach could be show to promote the expected qualitative effects of minimizing surprise in a simple setting.

---

> ### Author Response · Authors · 2021-11-13
> **Response to Reviewer doix**
>
> Respected Reviewer,
>
> Thank you for providing detailed feedback on the paper which is of utmost value to our work. We address your concerns (highlighted in __bold__) below and in the general response comment above.
>
> __I would expect some sort of qualitative evaluation showing that the agents in fact behave in a way to minimize surprise.__
> Thank you for pointing this out. We address this concern by providing a comprehensive evaluation of qualitative behaviors in Appendix E.1 and videos at the project website (also added to Abstract). Figures 5 and 6 present behaviors learned by EMIX and QMIX on the challenging so_many_baneling and 2s_vs_1sc tasks. While QMIX naively aims to maximize returns in the absence of surprise minimization, EMIX agents adapt as per the changing dynamics of the environment. For instance, in the so_many_baneling scenario consisting of 27 baneling opponents, EMIX agents learn an optimal strategy wherein the players continuously rearrange themselves at the corners of the bridge before approaching the enemy head-on. Note that this provides the players with additional cover, indicating that policies have been made aware of the incoming fast-paced assault. We have added this discussion as part of our results.
>
> __For instance in Table 1 it is difficult to tell which of the results are statistically significant.__
> We further evaluate the statistical significance of our methods in Appendix E.2 using the Mann-Whitney U test following the guildines of [1]. Each algorithm's 5 seeds are compared to that of the off-the-shelf QMIX baseline to obtain the $\mathcal{U}$ statistic. $\mathcal{U}$ here denotes a measure of statistical significance with higher values being desirable. As presented in Table 3, EMIX depicts a higher value of $\mathcal{U}$ in comparison to the most competitive SMiRL baseline. Additionally we expand our experiments to other Predator-Prey domains in Appendix A.3. The results presented here demonstrate a sound improvement over prior methods.
>
> __It isn't clear to me why you would want surprise minimization in multi-agent RL particularly__
> The need for surprise minimization in MARL arises from two sources (following Sections 1 and 2);
> (1) the abrupt dynamics of the MDP
> (2) a large portion of dynamical changes consisting of agent/opponent actions in the environment.
>
> Note that aspect (1) is common to single-agent surprise minimization as any agent can be uncertain about changes occuring in the environment [2]. Aspect (2), on the other hand, emphasizes the need for surprise minimization in MARL. In the case of increasing number of agents/opponents, various factors such as an exponential blowup of the action space [3] and approximation limitations of value factorization [4] take place. These inherently form a central part of environment dynamics and make agents uncertain about forthcoming changes. Surprise minimization, thus acts as a potential remedy to alleviate these instabilities and steer agents towards minimum energy niches.
>
> __What is different about multi-agent RL that would effect the sorts of behavior we would expect out of surprise minimization. I think this could be alleviated by an example showing how minimizing surprise in MARL produces desirable effects__
> Thank you for this suggestion. Following the previous concern, the primary difference between the two settings is the increased dynamical changes arising from agent/opponent interactions. This presents a dire need to tackle surprise as each agent, in addition to the environment dynamics, should also be aware of actions of other agents and opponents. We demonstrate this insight on a small Predator-Prey toy example in Appendix E.3 following your suggestion. Figure 7 compares QMIX with EMIX in the presence of increasing number of opponents. Due to the presence of surprise minimization, EMIX is found robust to scenarios with larger number of agents presenting a higher degree of uncertainty.
>
>
> __In the final paragraph of page 3, the $Q_a$ is said to be lower bounded by $Q^*$ when it is demonstrated to be upper bounded by $Q^*$__
> We apologize, this typing error has now been fixed.
>
> Please let us know if we have been able to address your concerns.
>
> [1]. Michael Lones, How to avoid machine learning pitfalls: a guide for academic researchers, 2021.
> [2]. Berseth et. al., SMiRL: Surprise Minimizing Reinforcement Learning in Unstable Environments, ICLR 2021.
> [3]. Mahajan et. al., Tesseract: Tensorised Actors for Multi-Agent Reinforcement Learning, ICML 2021.
> [4]. Rashid et. al., Weighted QMIX: Expanding Monotonic Value Function Factorisation for Deep Multi-Agent Reinforcement Learning, NeurIPS 2020.

---

> > ### Comment · Reviewer_doix · 2021-11-19
> > **Reply to Author Response**
> >
> > Thank you for your detailed response.
> >
> > The qualitative examples do clarify slightly, but it is difficult to distinguish the exhibited behaviors from the agent being more capable at the tasks or being in a different local minima.  In particular, it is not clear to me why taking cover before attacking is any less surprising than attacking directly?  The first seems like it could even be more surprising since there are multiple phases.
> >
> > I do not understand the explanation of the other task behavior either:
> > >The enemy, having a long tentacle, chooses
> > to attack any one of the agents randomly. This indicates that the agents face a greater degree of
> > surprise with no prior knowledge of the enemy’s movement.
> >
> > I don't understand why this should let us predict that the surprise-minimizing agents would approach from opposite sides.  Does approaching the enemy from opposite sides make the attacker attack less randomly?
> >
> > >For instance in Table 1 it is difficult to tell which of the results are statistically significant.
> >
> > More to the point here, for the majority of the results where your method is bolded as best in Table 1, the standard deviation EMIX and QMIX overlap.  It is very difficult to tell visually by looking at this table that this is the case, so the bolding scheme gives a false sense of the results being more significant than they are.  Moreover, if the improvement is that marginal it is hard to justify the added complexity.  I think that this approach could be justified qualitatively even without these quantitative results, but that qualitative argument would have to be clear and have clear use cases.
> >
> >
> > >  Surprise minimization, thus acts as a potential remedy to alleviate these instabilities and steer agents towards minimum energy niches.
> >
> > It would be great if you could offer a specific situation in which this can be shown to play out formally, in which your method can be shown to find that particular minimum energy niche.  Otherwise this argument seems abstract and it is difficult to tell if this is an active issue in any particular setting or if that issue has been solved by your approach.
> >
> >
> > >We demonstrate this insight on a small Predator-Prey toy example in Appendix E.3 following your suggestion. Figure 7 compares QMIX with EMIX in the presence of increasing number of opponents. Due to the presence of surprise minimization, EMIX is found robust to scenarios with larger number of agents presenting a higher degree of uncertainty.
> >
> > I appreciate these follow up experiments, but I'm not sure it addresses my confusion.  I do not follow the intuition that surprise minimization would certainly help in the predator-prey game, and to understand the strengths and limitations of the approach it is important to understand what properties of the environment you need in order for the method to work.  I think it would be very informative to readers to walk through a single example step by step to see what the surprise minimizing strategy actually is, so that you can argue that is good in a concrete instance.  Without this level of specificity it is hard to determine if the method is having the effect through the desired mechanism.

---

> > > ### Author Response · Authors · 2021-11-20
> > > **Response to Followup**
> > >
> > > Thank you for the followup questions. We further clarify your concerns in detail-
> > >
> > > __In particular, it is not clear to me why taking cover before attacking is any less surprising than attacking directly?__
> > > Finding cover before carrying out an assault minimizes damage to agents. Additionally, the cover acts as a safety spot where agents can stand while other agents attack the enemy. The reason why EMIX presents this as a surprise minimizing behavior is due to the definition of the task. Note that the enemy approaches from the front which leaves the agent team in the open. An off-the-shelf MARL agent (QMIX) decides to meet the enemy head on which results in more frequent damage. EMIX policies, on the other hand, execute a surprise-robust behavior by moving the agent away from the incoming attack (the surprising element). After finding cover, agents carry out the assault to maximize expected returns. Not only this strategy results in lower damage, but also yields higher success rates (c.f Table 1).
> > >
> > > __Does approaching the enemy from opposite sides make the attacker attack less randomly?__
> > > The task explanation has been updated as follows- "The enemy, having a long tentacle *pointing to the front*, chooses to attack any one of the agents randomly *in front of it*. Additionally, the tentacle has a fixed length and cannot extend beyond this range." Here we emphasize that the enemy can only attack the agents in front of it as the fixed length tentacle points to the front. With that said, the surprise-robust policy aims to evade random intermittent attacks (the surprising element) by going around to secure a safe position. Following a safe location, the agent attacks the enemy to inflict damage. Thus, approaching the enemy from opposite side does not make the attacks less random. This would only occur if the dynamics of the MDP were to be changed. Instead, approaching the enemy from opposite side helps the agent evade the incoming surprise which is exactly what is expected of the surprise minimizing policy.
> > >
> > > __the standard deviation EMIX and QMIX overlap.__
> > > We apologize. To avoid any further confusion, we have also bolded QMIX in settings where this is the case. We further direct readers to Appendix E.2 for a thorough analysis.
> > >
> > > __It would be great if you could offer a specific situation in which this can be shown to play out formally, in which your method can be shown to find that particular minimum energy niche.__
> > > We answer your concern from both perspectives-
> > >
> > > __Empirical-__ We do build our method on top of the QMIX framework which demonstrates approximation limitations of value factorization [1]. Note that we do not claim improving the value factorization process in any way as that is not the primary objective of our study. However, this process acts as one of the reasons for inducing surprise in the presence of large opponents.
> > >
> > > __Theoretical-__ More formally, we do provide a convergence guarantee for finding minimum energy niches (in Appendix C). The guarantee holds for *any* MARL as well as single-agent scheme which employs surprise minimization in the function approximation setting. This theoretical result in itself highlights the energy-based surprise minimization as a theoretically powerful framework for learning surprise-robust policies while maximizing expected returns.
> > >
> > > __I do not follow the intuition that surprise minimization would certainly help in the predator-prey game.__
> > > The predator-prey game presents varying number of opponents which increase the uncertainty of agents in the environment [2]. These agents are the only source of surprise as the MDP dynamics are held fixed. When the number of agents are varied, EMIX is found more robust to uncertain changes of opponents. This highlights that robustness to a large degree of uncertainty is what is expected from the surprise-robust policy. Utilizing this understanding for Table 1 and our qualitative analysis (Appendix E.1), we see that this is exactly reflected in policy's performance. In Table 1, EMIX improves over QMIX and SMiRL-QMIX in scenarios where the degree of uncertainty is high. In Appendix E.1, EMIX is found to qualitatively address this high uncertainty using surprise-robust behaviors.
> > >
> > > On the step-by-step walkthrough, we do note that building intuition using the toy task would be a suitable way to organize insights. However, we highlight the theoretical nature of our study and the required assumptions/guarantees.We believe that emphasizing on the theoretical aspects first is essential for its practical realization. At the same time, a small example which demonstrates key insights of our approach is a great addition. We sincerely thank you for this suggestion.
> > >
> > > [1]. Rashid et. al., Weighted QMIX: Expanding Monotonic Value Function Factorisation for Deep Multi-Agent Reinforcement Learning, NeurIPS 2020.
> > > [2]. Berseth et. al., SMiRL: Surprise Minimizing Reinforcement Learning in Unstable Environments, ICLR 2021.

---

> > > > ### Comment · Reviewer_doix · 2021-11-26
> > > > **Response to "Response to Followup"**
> > > >
> > > > > the task explanation has been updated as follows- "The enemy, having a long tentacle pointing to the front, chooses to attack any one of the agents randomly in front of it. Additionally, the tentacle has a fixed length and cannot extend beyond this range." Here we emphasize that the enemy can only attack the agents in front of it as the fixed length tentacle points to the front. With that said, the surprise-robust policy aims to evade random intermittent attacks (the surprising element) by going around to secure a safe position.
> > > >
> > > > I see.  The thing that makes "surprise minimization" work for these tasks is that unexpected events are usually bad (typically some sort of attack), thus minimizing surprise is a good reward-shaping term which will make the agents more likely to learn good strategies.  This makes sense to me, and also highlights the limitations of the approach.  Essentially the approach works if unexpected events are bad, and doesn't if unexpected events are good.  For instance, if posed with a free lottery ticket, the agent would not want to check if it has a winning number because it would be surprised if it were winning.
> > > >
> > > > This is a fine limitation to have, but it should be made clear and emphasized early on so it is clear that this approach is reliant on unexpected events being bad.  It would also be good to explain concretely how this assumption plays out in the environments: getting shot is unexpected and bad in the first env, and getting attacked is unexpected and bad in the second env.
> > > >
> > > > >It would be great if you could offer a specific situation in which this can be shown to play out formally, in which your method can be shown to find that particular minimum energy niche.
> > > > We answer your concern from both perspectives-
> > > >
> > > > >Empirical- ....
> > > >
> > > > >Theoretical- ...
> > > >
> > > > Thank you for the comments on how we can ensure that it finds an equilibrium.  However I thought that you were claiming something more when you mentioned it as a way of "potential remedy to alleviate these instabilities" with respect to MARL.  When I imagine these instabilities, it is usually things like coordination problems, finding cooperative plans, and I was reading the paper and responses to be suggesting that surprise minimization was a natural way to promote good group-dynamics within the multi-agent system.  It is not clear to my why this would be the case, even given that the agents find an equilibrium, I do not see why we should expect it to be better than before.  Using the prior arguments I would guess an argument for this may factor through "surprises in multi-agent systems are usually bad", but I don't know why that would be the case.  If there is a clear function this surprise minimization would serve for group-dynamics that is not clear in the current writing.
> > > >
> > > > Overall this leaves me at a similar place to Reviewer 8Wiu, with their first weakness:
> > > > >Although the authors claimed that the proposed learning framework is for MARL, the whole theoretical framework is weakly connected to the context of MARL (in decentralized manner).
> > > >
> > > > However, given that  explanations of the qualitative behaviors make sense, and the aim of this approach is to produce qualitatively different behaviors, I will be adjusting my score to weak accept.  However, I would expect a clear explanation that this approach relies on the idea that the designer knows ahead of time that "unexpected events are usually bad", to emphasize that this isn't universally true, but to provide simple motivations where we should expect generally that "unexpected events are usually bad".  The example where the only unexpected events that exist are attacks, accidents, or other system failures makes sense to me.  I think adding these and emphasizing them early is critical for the reader to understand the utility and limitations of the approach.

---

> > > > > ### Author Response · Authors · 2021-11-26
> > > > > **Response**
> > > > >
> > > > > Thank you for raising your score and supporting acceptance for our work. We are happy to know that the conerns regarding qualitative analysis were addressed.
> > > > >
> > > > > We also thank you for sharing your detailed insights and intuition on good and bad surprise. This is exactly in line with current works on surprise minimization [1], with some being at this conference [2]. Our work follows the precept of these works with the assumption that surprise affects the Multi-Agent population in a negative way, e.g- fire attacks, sudden appearances, distant shots, etc. While the setting of good surprise in literature remains less understood, it definitely highlights future avenues for our work.
> > > > >
> > > > > Following the discussion, we will improve the revised/final verion of the paper to incorporate your suggestions. Specifically, we aim to add the following-
> > > > > + Additional discussion on positive and negative surprise and their effects
> > > > > + Additional functions of surprise minimization and its applicability to other learning problems
> > > > > + Explicit mention of the kind of surprise following prior works with clarity on its negative effects
> > > > >
> > > > > [1]. Berseth et. al, SMiRL: Surprise Minimizing Reinforcement Learning in Unstable Environments, ICLR 2021.
> > > > > [2]. Anonymous, Explore and Control with Adversarial Surprise, Under Review at ICLR 2022.

---

> ### Author Response · Authors · 2021-11-18
> **Discussion**
>
> Kindly let us know if our response below addressed your concerns. We will be happy to answer if there are additional issues/questions.

---

### Author Response · Authors · 2021-11-13
**General Response to Reviewers**

Respected Reviewers,

Thank you for providing detailed feedback on our work. Since the main concerns of reviewers focus on the empirical evaluation and technical discussion of surprise , we have updated the manuscript with the following additions highlighted in $\color{blue}{\text{blue}}$-
+ Additional experiments on Predator-Prey benchmark over 5 random seeds in Appendix E.3
+ Statistical significance of our results on SMAC benchmark in Appendix E.2
+ Qualitative results of agent behaviors in Appendix E.1 with videos at the anonymous project website
+ Concise discussion on relation to maximum entropy framework in Appendix B.1
+ Extension of discussion on convergence and role of temperature in Appendix C
+ Discussion on selection and tuning of $\beta$ in Appendix D.3
+ Discussion on computation of deviation and surprise estimates in Appendix D.1
+ Clarification of notations and definitions in Section 3.1
+ Correction of minor typing errors and Fig. 1 in Section 4.1
+ $\color{red}{\text{[NEW]}}$ Additional ablations in Appendix E.4
+ $\color{red}{\text{[NEW]}}$ Additional qualitative demos on project website

---

### Author Response · Authors · 2021-11-22
**Additional Ablations and Qualitative Demos**

Respected Reviewers,

In addition to the below changes, we have further added ablations for temperature parameter $\beta$ (Appendix E.4) and qualitative demos on the anonymous project website (linked in Abstract). These changes are marked as $\color{red}{\text{[NEW]}}$ in our general response comment below.

Once again, thank you for evaluating our work and providing valuable feedback!

---

### Decision · Program_Chairs · 2022-01-20

**Decision:**

Reject

**Comment:**

The paper explores surprise minimization in multi-agent learning by using free energy across all agents in a multi-agent system. A temporal EBM represents an estimate of surprise which is minimized over the joint agent distribution. Empirical studies on the proposed method are conducted. This paper builds in an interesting direction around surprise minimization in multi-agent learning by using the energy-based framework, but the presentation of the method seems to need more efforts to be improved to avoid confusion.

The discussion between authors and reviewers is summarized below: The major concerns of Reviewer doix include that: (i) the empirical results are not compelling, (ii) qualitative results are missing, and (iii) the motivation of surprise minimization in multi-agent RL is unclear. After the rebuttal, the authors addressed the concerns of Reviewer doix, who changed his/her score from 5 to 6. The major concern of Reviewer zFND comes from the understanding and justification of the paper. After the rebuttal, the concerns of Reviewer zFND  have been partially addressed by clarification on what measures of surprise can be used and how these would be estimated. Reviewer zFND eventually changed his/her score from 5 to 6.  Also, most of the concerns about theory and experiments from Reviewer 8Wiu have been addressed after the rebuttal. Reviewer 8Wiu accordingly changed the rating from 5 to 6. Reviewer g9cM is still not satisfied with the authors' answers, and his/her concerns regarding some technical issues remain and points out that the current paper has many inconsistencies across the writing that make it hard to evaluate the soundness and correctness of the results.

After the rebuttal, the author successfully addressed most of the concerns from 3 of 4 reviewers, but the overall rating of the paper is on a borderline level. Given the fact that the paper still has some unaddressed concerns from Reviewer g9cM, and other reviewers actually do not champion the paper. The AC tends to recommend rejecting the paper at the current stage. AC urges the authors to improve their paper by including all the suggestions provided by the reviewers, and then resubmit it to a future venue.